# Combined KRAS-MAPK pathway inhibitors and HER2-directed drug conjugate is efficacious in pancreatic cancer

Ashenafi Bulle[1,6], Peng Liu[1,2,6], Kuljeet Seehra[1], Sapana Bansod[1], Yali Chen[1], Kiran Zahra[1], Vikas Somani ⬤[1], Iftikhar Ali Khawar[1], Hung-Po Chen[1], Paarth B. Dodhiawala[1], Lin Li[1], Yutong Geng[1], Chia-Kuei Mo[1], Jay Mahsl ⬤[1], Li Ding ⬤[1], Ramaswamy Govindan ⬤[1], Sherri Davies ⬤[1], Jacqueline Mudd ⬤[3], William G. Hawkins[3], Ryan C. Fields ⬤[3], David G. DeNardo ⬤[1], Deborah Knoerzer[4], Jason M. Held ⬤[1], Patrick M. Grierson[1], Andrea Wang-Gillam[1], Marianna B. Ruzinova[5] & Kian-Huat Lim ⬤[1] ✉

Targeting the mitogen-activated protein kinase (MAPK) cascade in pancreatic ductal adenocarcinoma (PDAC) remains clinically unsuccessful. We aim to develop a MAPK inhibitor-based therapeutic combination with strong pre-clinical efficacy. Utilizing a reverse-phase protein array, we observe rapid phospho-activation of human epidermal growth factor receptor 2 (HER2) in PDAC cells upon pharmacological MAPK inhibition. Mechanistically, MAPK inhibitors lead to swift proteasomal degradation of dual-specificity phosphatase 6 (DUSP6). The carboxy terminus of HER2, containing a TEY motif also present in extracellular signal-regulated kinase 1/2 (ERK1/2), facilitates binding with DUSP6, enhancing its phosphatase activity to dephosphorylate HER2. In the presence of MAPK inhibitors, DUSP6 dissociates from the protective effect of the RING E3 ligase tripartite motif containing 21, resulting in its degradation. In PDAC patient-derived xenograft (PDX) models, combining ERK and HER inhibitors slows tumour growth and requires cytotoxic chemotherapy to achieve tumour regression. Alternatively, MAPK inhibitors with trastuzumab deruxtecan, an anti-HER2 antibody conjugated with cytotoxic chemotherapy, lead to sustained tumour regression in most tested PDXs without causing noticeable toxicity. Additionally, KRAS inhibitors also activate HER2, supporting testing the combination of KRAS inhibitors and trastuzumab deruxtecan in PDAC. This study identifies a rational and promising therapeutic combination for clinical testing in PDAC patients.

More than 90% of pancreatic ductal adenocarcinoma (PDAC) cases harbor activating mutations in the KRAS (Kristen's rat sarcoma viral oncogene homolog (KRAS) gene[1]. KRAS mutant proteins exert their oncogenic functions by engaging multiple downstream signaling cascades, including the RAF-MEK-ERK mitogen-activated protein kinase (MAPK) and PI3K-AKT-mTOR pathways. The MAPK pathway is considered one of the most critical therapeutic targets based on numerous preclinical studies[2]. However, MAPK inhibitors, in combination with targeted agents including PI3K inhibitors or chemotherapy, have lacked efficacy in clinical trials and have been shown to have significant

side effects[3–5]. Our study aimed to develop MAPK inhibitor-based therapeutic combinations that are based on solid scientific rationale. We use patient-derived xenografts (PDXs), which are the most clinically relevant models to study non-immunologic mechanisms of drug resistance and to test therapeutic combinations, as they faithfully recapitulate the genetic complexity and drug responses of primary human tumours[6].

Here, we show that MAPK inhibition results in phospho-activation of human epidermal growth factor receptor 2 (HER2). Mechanistically, the TEY motif at the C-terminus of HER2 binds and is dephosphorylated by dual-specificity phosphatase 6 (DUSP6), and the stability of DUSP6 protein is mediated by the E3 ligase Tripartite Motif Containing 21 (TRIM21). Treatment with MAPK inhibitors dissociates TRIM21 from DUSP6, causing it to be polyubiquitinated and degraded, thereby leaving HER2 in a sustained phosphorylated state. While the combination of small molecule MAPK and HER inhibitors is modestly effective, the addition of cytotoxic chemotherapy is required to achieve clinically meaningful efficacy. Alternatively, MAPK inhibitors in combination with DS-8201a, which is a trastuzumab antibody conjugated with a cytotoxic payload, deruxtecan, were highly effective, resulting in complete tumour regression in multiple PDX models. Lastly, KRAS inhibitors also upregulate HER2 phosphorylation, urging clinical testing of the combination of KRAS inhibitor plus DS-8201a for PDAC patients.

## Results

### Targeting the MAPK pathway is inadequate and compromised by HER2 activation

To date, combination chemotherapies remain the key treatment component for PDAC, but the objective response rate is less than 40% and typically short-lived, underlying the need to develop strategies that can augment treatment responses. Gemcitabine at concentrations up to 20 μM which covers the half-maximal inhibitory concentration (IC$_{50}$) for most PDAC cell lines[7,8], dose-dependently induced cleaved caspase-3 and simultaneously phospho-activated ERK1/2 kinases (p-ERK1/2) and their canonical substrate p90RSK (p-p90RSK) to various degrees in KRAS-mutant PDAC cells, including patient-derived cell lines (PDCLs, Fig. 1a, Supplementary Fig. 1a, b). Other chemotherapeutic agents used in PDAC treatment, including oxaliplatin, irinotecan, 5-fluorouracil (5-FU) and paclitaxel, also robustly activated ERK1/2, but not the PI3K-AKT cascade (Fig. 1b). PDAC xenograft tumours treated with gemcitabine showed a markedly stronger p-ERK1/2 signal by immunohistochemical (IHC) staining (Supplementary Fig. 1c). Furthermore, analyses of TCGA database showed a very strong and significant positive correlation between MAPK and two independent gemcitabine resistance gene expression signatures[9,10] ($p$ ~ 0, Pearson coefficients >0.8, Fig. 1c). These data suggest MAPK activation is a stress response that may confer chemoresistance. However, a phase II clinical study showed that addition of trametinib, a MEK inhibitor, to gemcitabine did not improve treatment response or survival in PDAC patients[11], potentially due to mechanisms that reactivate downstream ERK kinases[12,13]. Ulixertinib is a first-in-class ERK1/2 inhibitor with single-agent activity in MAPK pathway-mutant cancers demonstrated in a phase 1 clinical trial[14,15]. We found that ulixertinib alone showed dose-dependent induction of cleaved caspase-3, suppression of p-p90RSK, and a paradoxical increase in p-ERK1/2 (Supplementary Fig. 1d, e), as published[16,17]. Gemcitabine and ulixertinib showed synergism in suppressing PDAC cell viability, as defined by combination indices (CI, by the Chou-Talalay method[18]) of <0.9 (Supplementary Fig. 1f). The addition of ulixertinib to gemcitabine induced more apoptosis according to Annexin-V staining using flow cytometry (FACS) analysis compared to either agent alone (Supplementary Fig. 1g). In a murine xenograft experiment using KRAS-mutant PDCLs (Pa01c and Pa02c), the triplet of ulixertinib, gemcitabine and paclitaxel was more effective in suppressing, but did not regress, tumour growth

(Fig. 1d). A recent phase 1b clinical trial combining ulixertinib, gemcitabine, and nab-paclitaxel conducted at Washington University in St. Louis was prematurely terminated due to poor patient tolerability, although ulixertinib was able to downregulate KRAS-dependent gene signatures in tumour samples[5]. This setback led us to explore other MAPK inhibitor-based therapeutic strategies that have strong preclinical efficacy.

To identify actionable adaptive responses to MAPK inhibition, we performed reverse-phase protein array (RPPA) on KRAS$^{G12D}$-mutant Pa01c and HPAC cells treated with trametinib or ulixertinib for 24 h (Fig. 1e). We focused on changes that were altered more than two-fold. As expected, both inhibitors downregulated the phosphorylation of ERK substrates S6 and S6K[19], and cell cycle proteins such as phosphorylated Rb, Cdc42[20,21], Aurora A, and B[22]. Dual-specificity phosphatases 4 and 6 (DUSP4, DUSP6), which are negative regulators of ERK1/2[23], were also downregulated. Upregulated markers included phospho-HER2, total HER3, Matrix Metallopeptidase 14 (MMP14), epithelial membrane antigen (EMA, a.k.a Mucin1 or MUC1), Interferon Regulatory Factor 1 (IRF-1), and pro-apoptotic Bcl-2-like protein 11 (BCL2L11 or BIM, Fig. 1f). Phospho-ULK1(S757) was elevated in Pa01c, consistent with autophagy induction as previously published[24,25]. Feedback activation of HER signaling is a known mechanism of resistance to KRAS and MAPK pathway inhibitors in colon and pancreatic cancers[16,26,27]. Increased expression of HER1/EGFR, HER2, and HER3 by immunohistochemistry (IHC) has been reported in up to ~70%[28,29], ~60%[30,31], and ~24%, respectively[32] in PDAC. Western blotting showed that PDAC cell lines displayed various levels of total and phosphorylated HER2 (Supplementary Fig. 1h). In untransformed 293 T cells, which have low basal HER1–3 expression, ectopic expression of HER2, and to a lesser degree HER1, robustly enhanced p-ERK1/2, but this was abrogated by Afatinib (Fig. 1g). HER3 overexpression had no detectable impact on p-ERK1/2, consistent with it being a pseudokinase that needs to heterodimerize with HER1 or HER2 for signaling[33]. Interestingly, HER2 overexpression downregulated DUSP6, but not DUSP4, suggesting a mechanistic link between HER2 and DUSP6. We confirmed the RPPA results that ulixertinib and trametinib increased the phosphorylation of HER2 at residues Y1248, Y1221/1222, and Y877, but not at T686 and Y1196, as well as the phosphorylation of HER3 at Y1197, which mediates HER2/HER3 heterodimerization and activation of the PI3K pathway[34]. We also observed increased total HER2 and HER3 (Fig. 1h), as well as phospho-AKT as published[16]. Consistently, ulixertinib-treated PDX tumours displayed increased HER2 and p-ERK1/2, and decreased DUSP4 and DUSP6 IHC staining (Supplementary Fig. 1i).

To determine the impact of the significantly upregulated markers from RPPA, we treated seven PDAC cell lines with ulixertinib in combination with afatinib (HER inhibitor), Ro 28–2653 (MMP inhibitor), GO-201 (EMA/MUC1 inhibitor), or ruxolitinib (JAK inhibitor to block IRF-1 activation) and performed Annexin-V staining by FACS to assess apoptosis. Of these agents, only afatinib showed consistently greater pro-apoptotic effects when combined with ulixertinib or trametinib in all seven cell lines tested (Supplementary Fig. 2a, b). To corroborate the Annexin-V data, we performed drug combination experiments and found that afatinib was the only agent that consistently showed synergism with trametinib or ulixertinib in suppressing the growth of these PDAC cell lines (Supplementary Fig. 2c).

To dissect the role of each HER family member in MAPK inhibition, we knocked down ERBB1/2/3 each using two small hairpin (sh) RNAs in Pa01c and HPAC cells. ERBB2-silenced cells were the most sensitive to inhibition by ulixertinib or trametinib, as shown by the decrease in IC$_{50}$ values (Supplementary Fig. 2d). Furthermore, high expression of ERBB2 mRNA was associated with poor relapse-free survival (RFS) and overall survival (OS; Supplementary Fig. 2e). These data suggest that HER2 signaling is a resistance mechanism to MAPK inhibition.

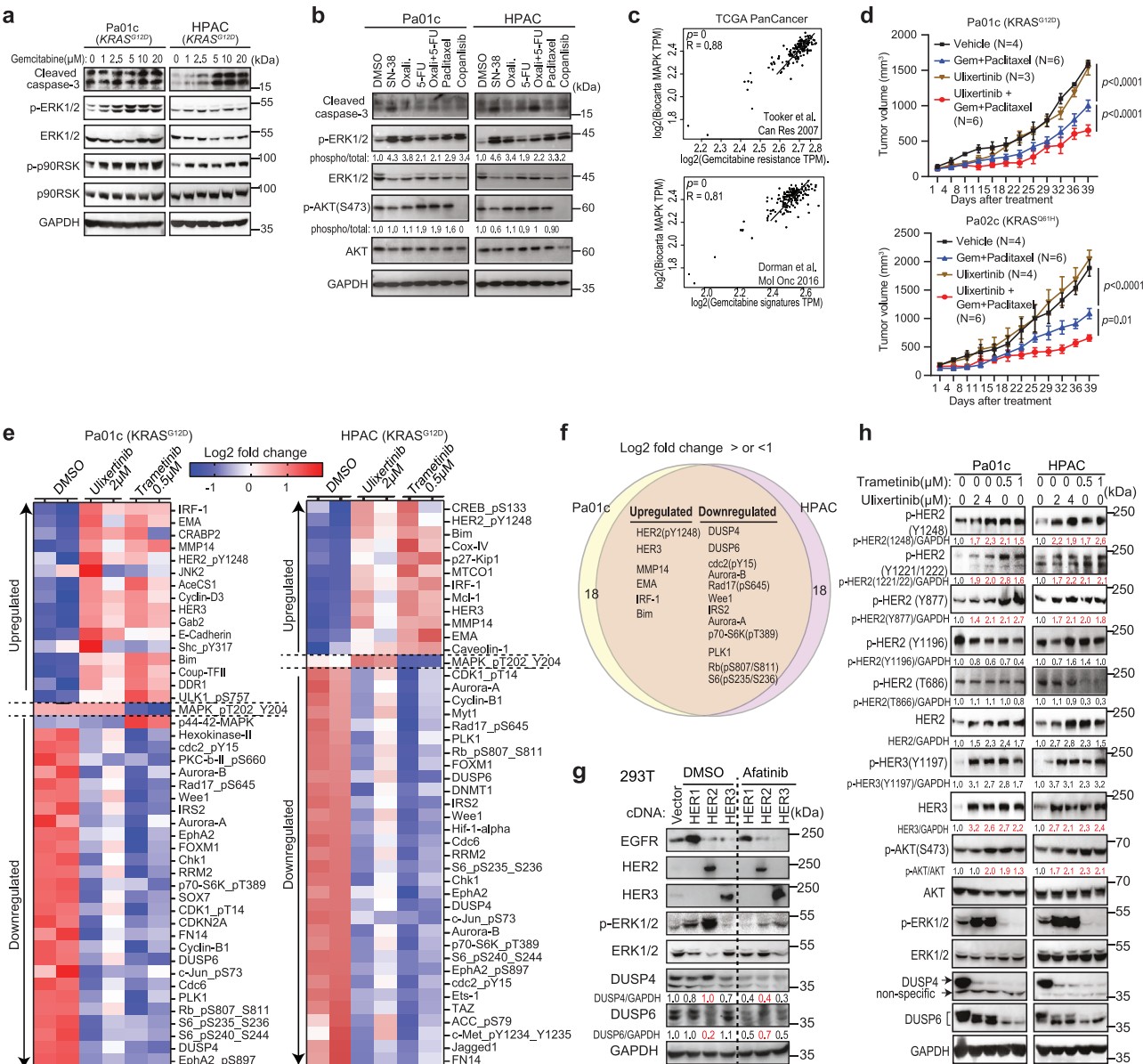

**Fig. 1 | Targeting the MAPK pathway is inadequate and compromised by HER2 activation. a** Western blots showing dose-dependent increases in phosphorylated ERK1/2, p90RSK, and cleaved caspase-3 levels in PDAC cells treated with the indicated concentrations of gemcitabine for 24 h. **b** Western blots showing changes in p-ERK1/2, p-AKT(S473), and cleaved caspase-3 levels after treatment with the indicated agents for 24 h. **c** Correlation plots with Pearson coefficients (R) showing strong positive correlation between MAPK and two independent gemcitabine resistance signatures in PDAC samples from TCGA PanCancer database. **d** Growth kinetics of subcutaneous Pa01c and Pa02c xenograft tumours treated as indicated when the tumour volume reached ~100 mm$^3$. Data are presented as mean ± SEM. *P*-values were calculated using two-way ANOVA with Tukey's multiple comparison

test. **e** Heatmap of RPPA data showing significantly upregulated and downregulated markers in Pa01c and HPAC cells treated with ulixertinib or trametinib for 24 h. **f** Venn diagram showing the shared changes in both cell lines. Only markers showing a Log2 fold change or <-1 or >1 are illustrated. Data are presented as the mean ± SEM of two biological samples. **g** Western blots showing changes in p-ERK1/2, DUSP4, and DUSP6 levels in 293 T cells transfected with HER1, HER2, or HER3 for 36 h and then treated with DMSO or afatinib for 16 h. **h** Western blots showing changes in different phosphorylated-HER2 signals, p-HER3, p-ERK1/2, p-AKT, DUSP4, and DUSP6, in Pa01c and HPAC cells treated with ulixertinib or trametinib for 24 h. **a**, **b**, **g**, **h** were conducted two times, and one set of data was presented. Source data are provided in Source Data file.

## Proteasomal degradation of DUSP6 sustains HER2 phospho-activation

Our RPPA and Western blot data demonstrated downregulation of DUSP4 and DUSP6 following MEK or ERK inhibition (Fig. 1e, h). DUSPs are dual-specificity MAP kinase phosphatases (MKPs) responsible for dephosphorylating tyrosine, serine, and threonine residues of different MAPK members[35]. Among them, DUSP-6, -7, and -9 are cytoplasmic MPKs that dephosphorylate ERK, while DUSP4 is an inducible nuclear MKPs that also includes DUSP1 and DUSP5[36]. Therefore, we hypothesized that the downregulation of DUSP4/6 may contribute to HER2

phosphorylation. First, we found that trametinib or ulixertinib downregulated DUSP6 and DUSP4, but not the other DUSP members in PDAC and 293 T cells (Fig. 2a, Supplementary Fig. 2f). Following ulixertinib treatment, DUSP4 and DUSP6 protein levels started to significantly decrease in Pa01c cells as early as 3 and 6 h, respectively (Supplementary Fig. 2g), suggesting that protein degradation is the main mechanism of downregulation. Indeed, ulixertinib treatment resulted in robust polyubiquitination of both DUSP4 and DUSP6 (Fig. 2b). Downregulation of DUSP4 and 6 was partially blocked when Pa01c and HPAC cells were co-treated with the proteasome inhibitor

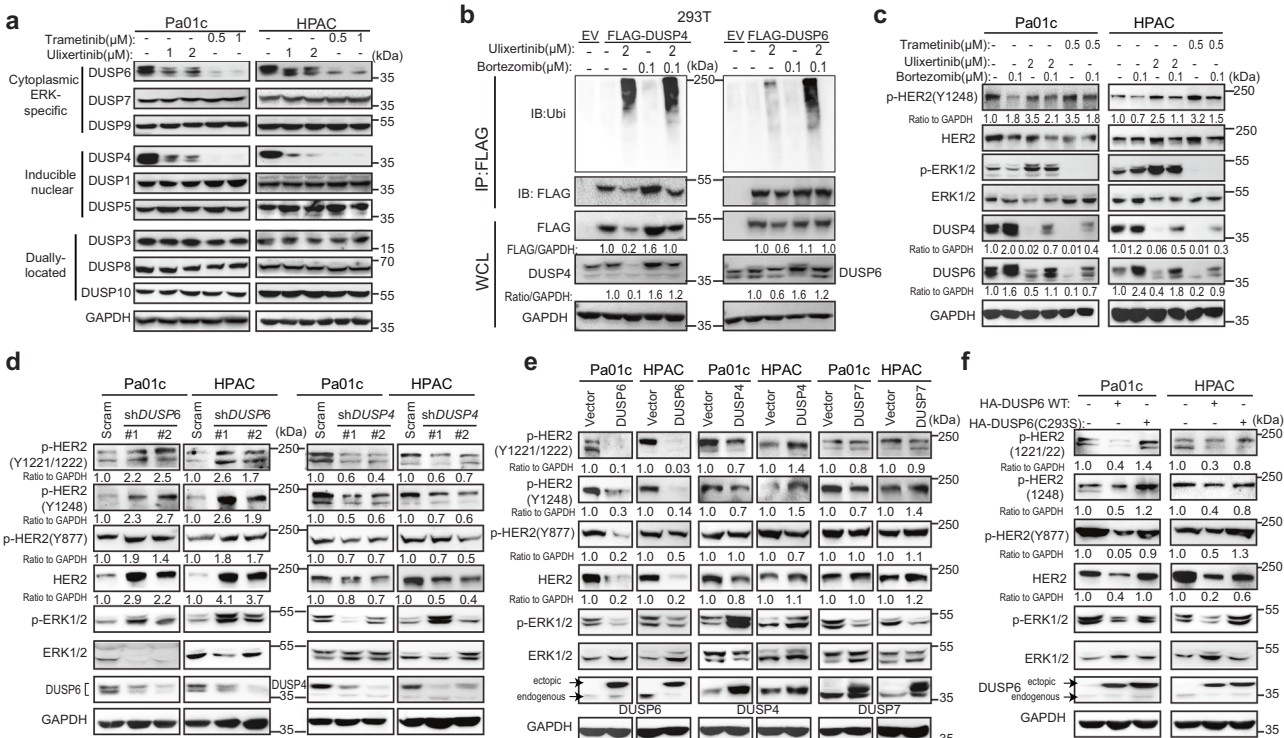

**Fig. 2 | Proteasomal degradation of DUSP6 sustains HER2 phospho-activation.**
**a** Western blots showing changes in the indicated DUSPs following treatment with trametinib or ulixertinib overnight (-16 h) in the two different PDAC lines.
**b** Immunoprecipitation (IP) experiment showing polyubiquitination of stably expressed FLAG-tagged DUSP4 and DUSP6 in 293 T cells following 16 h treatment with ulixertinib followed by co-treatment with DMSO or bortezomib for 6 h.
**c** Western blots showing changes in p-HER2 (Y1248), p-ERK1/2, DUSP4, and DUSP6 levels in Pa01c and HPAC cells treated with trametinib or ulixertinib for 16 h, followed by DMSO or bortezomib for 6 h. **d** Western blots showing changes in p-HER2

and p-ERK1/2 in Pa01c and HPAC cells stably expressing scramble control of two different shRNAs against *DUSP4* or *DUSP6*. **e** Western blots showing changes in p-HER2 and p-ERK1/2 in Pa01c and HPAC cells stably overexpressing an empty vector, DUSP6, DUSP4, or DUSP7. **f** Western blots showing changes in p-HER2 and p-ERK1/2 levels in Pa01c and HPAC cells stably expressing wild-type (WT) or enzymatically inactive (C293S) DUSP6. All experiments were conducted two times, and one set of data for each was presented. Source data are provided in Source Data file.

bortezomib (Fig. 2c). Notably, this restoration of DUSP4 or DUSP6 protein levels by bortezomib coincided with the attenuation of phospho-HER2 induced by trametinib or ulixertinib, providing further support for the idea that DUSP4 or DUSP6 may dephosphorylate HER2.

Next, we found that knockdown of DUSP6, but not DUSP4, resulted in increased phosphorylation of HER2 and as expected, ERK1/2 (Fig. 2d). Conversely, overexpression of DUSP6, but not DUSP4 or DUSP7, decreased the total and phosphorylated HER2 at Y1221/1222, Y1248, and Y877 (Fig. 2e). Wild-type, but not enzymatically inactive (C293S[37]) DUSP6 decreased phosphorylated HER2 and p-ERK1/2 (Fig. 2f). These data support DUSP6 as the phosphatase that mediates HER2 dephosphorylation.

**The TEY motif of HER2 promotes binding to DUSP6**
DUSP6 is best known as a direct phosphatase of ERK1/2. Phosphorylation of ERK2 at the T185 and Y187 in the TEY (codon 185–187) motif results in ~6-fold increased affinity of its common docking motif to DUSP6[38], which in turn dephosphorylates ERK2 in a stepwise manner[39]. By aligning the amino acid sequences of ERK1/2 and HER1–3 proteins, we found that HER2 also contained a TEY motif (codon 877–879, Fig. 3a). To determine whether this TEY motif mediates binding with DUSP6, we synthesized recombinant His-tagged wild-type (WT) or T185A/Y187A ERK2, WT or T885A/Y887A HER2 C-terminus (codon 676-end) peptides in BL21 bacteria and incubated them in vitro with bead-bound HA-tagged DUSP6 produced in Pa01c cells. The HA beads were then washed, and the bound proteins were analyzed by Western blotting. We found that DUSP6 binds robustly with WT and, to a much lesser extent, mutant ERK2 and HER2 (Fig. 3b). In a reverse experiment,

we incubated His-beads bound with recombinant His-tagged WT or mutant ERK2 or HER2 with HA-DUSP6 expressing Pa01c cell lysates, and similarly observed diminished binding of HER2 and ERK2 mutants to DUSP6 (Fig. 3c). To strengthen these findings, we treated purified FLAG-HER2 synthesized in 293 T cells with recombinant DUSP6 in vitro and showed a reduction of phospho-HER2 at Y877 (Fig. 3d). It is known that ERK2 binds and promotes the phosphatase activity of DUSP6[40], which in turn dephosphorylates ERK2. To determine whether similar phenomenon occurs with HER2, we performed in vitro phosphatase assay[41] using recombinant DUSP6, and found that treatment with purified HER2 increased the in vitro phosphatase activity of DUSP6 by ~3-fold (Fig. 3e). Together, these data suggest that the TEY motif of HER2 mediates binding with DUSP6 and promotes its phosphatase activity.

Next, we investigated the mechanism underlying the regulation of DUSP6 protein stability. We observed that HER2 overexpression, but not HER1 or HER3, downregulated DUSP6 (Fig. 1g). Furthermore, co-expression of HER1/HER2 or HER2/HER3, but not HER1/HER3, down-regulated DUSP6 (Supplementary Fig. 3a), strongly suggesting a bidirectional regulation between HER2 and DUSP6. Indeed, short-term (4 h) treatment with trametinib or ulixertinib prior to DUSP6 degradation resulted in a detectable increase in the interaction between endogenous HER2 and DUSP6 in PDAC cells, as measured by the Duolink® in situ proximity ligation assay (Fig. 3f). HER2 overexpression led to marked polyubiquitination of DUSP6, and this phenomenon was augmented in the presence of MEK or ERK inhibitors (Fig. 3g, Supplementary Fig. 3b). Therefore, MAPK inhibition increased the binding of HER2 to DUSP6, potentially leading to DUSP6 degradation. In turn,

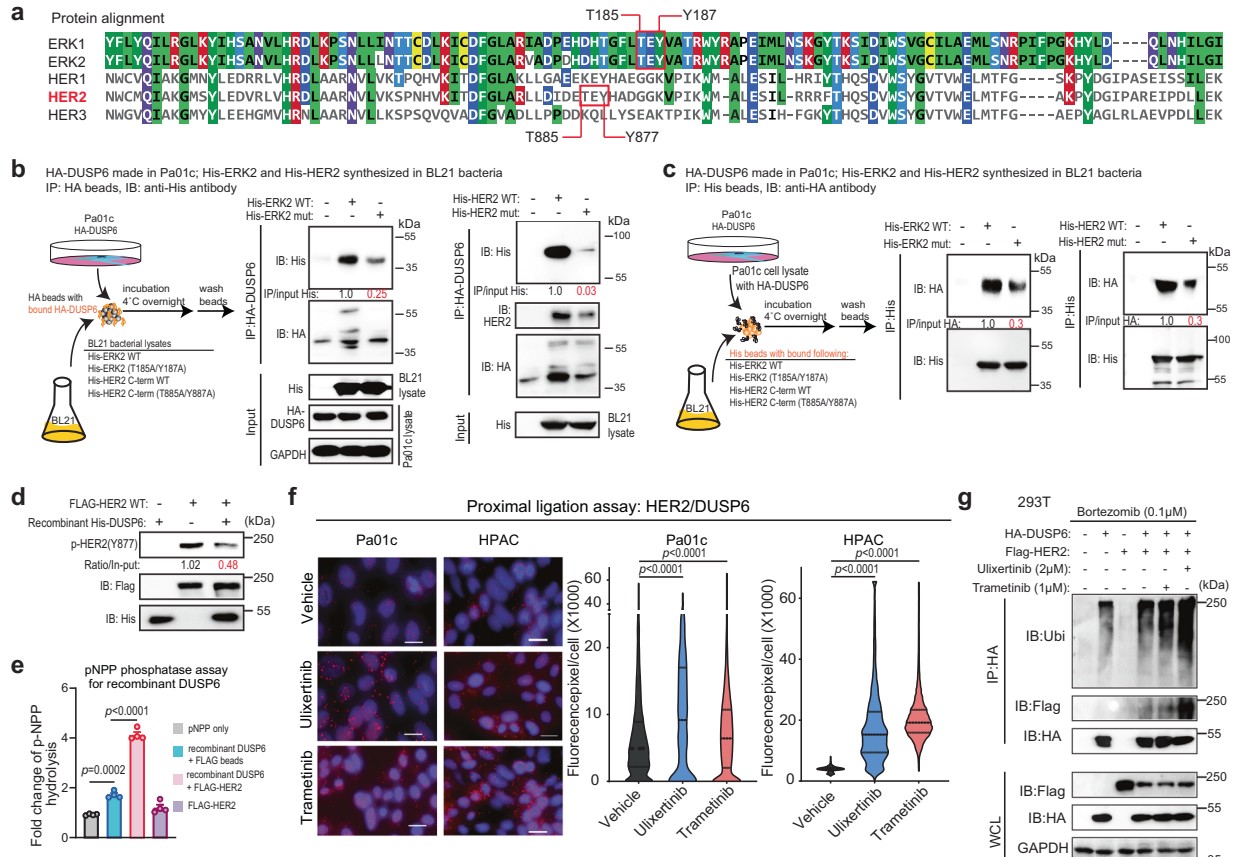

**Fig. 3 | The TEY motif of HER2 promotes binding to DUSP6. a** Alignment of amino acid sequences of the C-termini of ERK1, ERK2, HER1, HER2, and HER3, indicating the presence of the TEY motif in ERK1/2 and HER2. Schematic and results of IP experiments using HA-DUSP6 produced in human Pa01c cells and ERK2/HER2 variants using BL21 bacteria. IP was performed using HA beads (**b**) or His beads (**c**) to delineate the in vitro interaction between DUSP6 and HER2. **d** In vitro assay showing decrease phospho-HER2(Y877) of purified bead-bound FLAG-HER2 incubated with recombinant DUSP6 for 1 h. **e** In vitro pNPP phosphatase assay using recombinant DUSP6 and purified bead-bound FLAG-HER2 done in quadruplicates for 30 min at 37 °C in cell-free condition. The liberated p-nitrophenol which reflects DUSP6 phosphatase activity was measured at 405 nm using Biotek Synergy 2 spectrophotometer. **f** Representative immunofluorescence images and quantification of Duolink® proximity ligase assay identifying the interaction of endogenous HER2 and DUSP6 in Pa01c and HPAC cells treated with DMSO, ulixertinib 2 μM, or trametinib 1 μM for 4 h prior to degradation of DUSP6. Scale bar = 50 μm. For (**e**, **f**) data presented as mean ± SEM, *P* values were calculated by one-way ANOVA with Dunnet's multiple comparisons test or (**e**) two-tailed unpaired *t*-test (**f**). **g** IP experiment in 293 T cells showing polyubiquitination of ectopically expressed HA-tagged DUSP6 in the absence or presence of co-transfected FLAG-tagged HER2 as well as trametinib or ulixertinib treatment for 16 h. All cells were co-treated with bortezomib for at least 4 h prior to treatment with trametinib or ulixertinib to prevent the marked downregulation of DUSP6. Experiments in (**b**–**d**, **g**) were conducted two times, and one set of data for each was presented. Source data are provided in Source Data file.

lowered DUSP6 level results in increased HER2 phosphorylation, forming a positive feedforward loop.

## KRAS recruits E3 ligase TRIpartite Motif containing-21 (TRIM21) to regulates DUSP6 stability

Next, we sought to understand the mechanism governing DUSP6 degradation. In PDAC genetic mouse model, DUSP6 expression was upregulated in early preneoplastic lesions in response to oncogenic KRAS signaling[42]. In support, we found that expression of KRAS[G12V] also increased DUSP6, although it could still be partially degraded by trametinib or ulixertinib (Fig. 4a). Time-chase experiments using the protein translation inhibitor cycloheximide in isogenic 293 T cell lines showed that the decline of DUSP6 was slower in the presence of KRAS[G12V] (Fig. 4b). Conversely, silencing KRAS in KRAS[G12D]-mutant Pa01c cells caused DUSP6 to be less stable (Fig. 4c) and more readily degraded by ulixertinib (Fig. 4d). In support, ulixertinib-induced DUSP6 degradation was more pronounced when KRAS-mutant PDAC cells were co-treated with KRAS[G12D] inhibitor MRTX1133 in Pa01c and HPAC, or KRAS[G12C] inhibitor AMG-510 in MIA Paca-2 (Fig. 4e, f, Supplementary Fig. 4a). These data led us to hypothesize that KRAS oncoproteins participate in DUSP6 degradation.

To understand the mechanism, we immunoprecipitated FLAG-tagged KRAS[G12V] from 293 T cells and performed mass spectrometry to identify binding partners. To exclude non-specific hits, we used FLAG-tagged RALA[G23V], which shares ~85% amino acid sequence homology with KRAS[G12V] as a negative control. Reflecting the robustness of this data, RAF1 was enriched in KRAS[G12V] lysate whereas subunits of the exocyst complexes[43] were enriched in the RALA[G23V] lysate. We found that TRIpartite Motif containing-21 (TRIM21) as an E3 ligase that was enriched by more than 3-fold in KRAS[G12V] cells (Fig. 5a), leading us to test whether TRIM21 may degrade DUSP6. Western blotting confirmed that HA-tagged KRAS[G12V] indeed binds to endogenous TRIM21 and DUSP6. Surprisingly, these interactions were markedly diminished in the presence of trametinib or ulixertinib (Fig. 5b). Furthermore, time chase experiment with cycloheximide showed that overexpression of GFP-tagged TRIM21 rendered DUSP6 protein more stable (Fig. 5c). Additionally, wild-type, but not enzymatically inactive (C54Y[44]) TRIM21 diminished K48-polyubiquitination of DUSP6 (Fig. 5d) at baseline and in the presence of trametinib or ulixertinib (Fig. 5e, Supplementary Fig. 4b). Conversely, silencing TRIM21 by shRNAs decreased DUSP6 levels and correspondingly increased total and p-HER2 levels in PDAC cells (Fig. 5f). Phenotypically, TRIM21-silenced PDAC cells formed

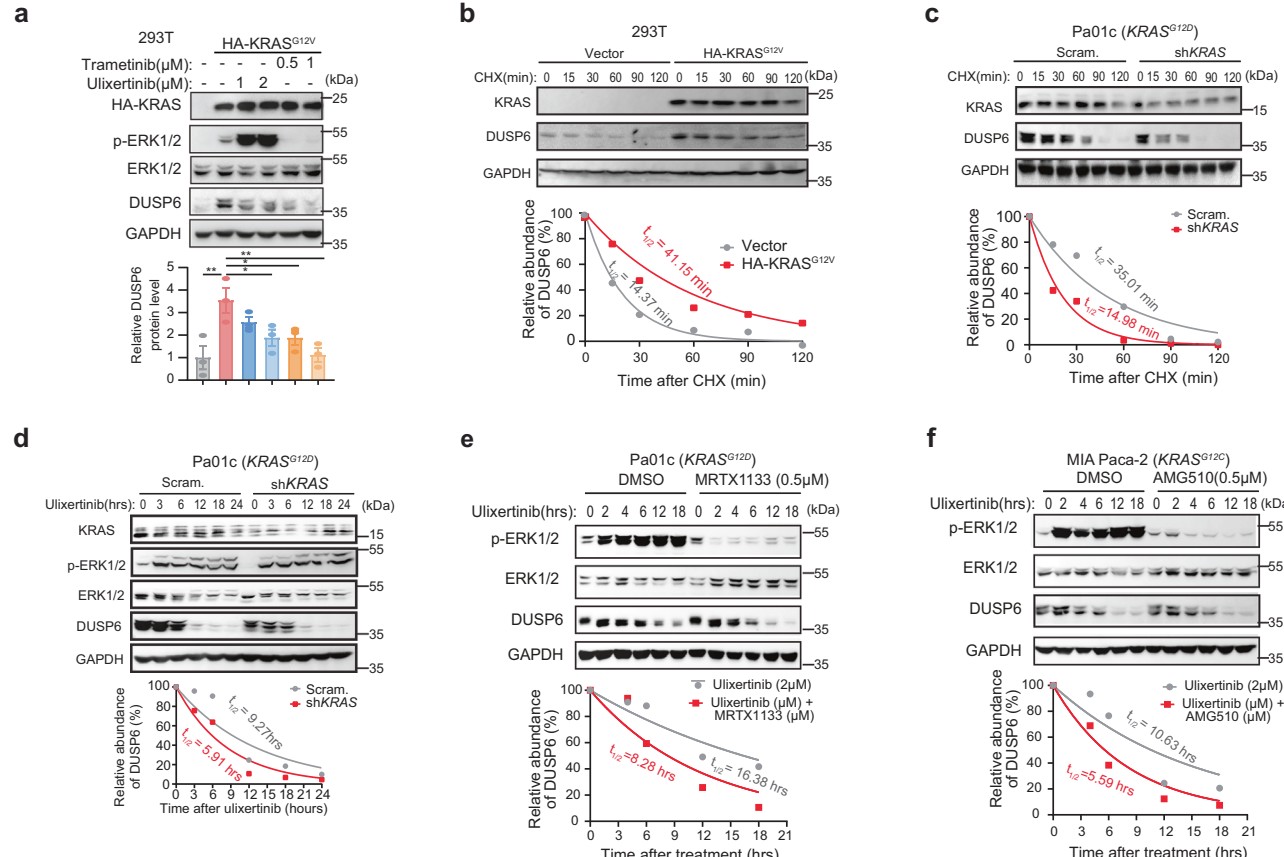

**Fig. 4 | KRAS regulates DUSP6 stability. a** Western blots showing changes in p-ERK1/2, DUSP4, and DUSP6 levels in 293 T cells stably expressing HA-KRAS$^{G12V}$ treated with trametinib or ulixertinib for 16 h. The bar graph includes quantification of three independent experiments. Data was presented as mean ± SEM. *P*-values were calculated using one-way ANOVA followed by Dunnet's multiple comparisons test, HA-KRAS vs control ($p = 0.0031$), HA-KRAS vs 2 µM ulixertinib ($p = 0.0474$), HA-KRAS vs 0.5 µM trametinib ($p = 0.0468$), HA-KRAS vs 1 µM trametinib ($p = 0.0045$). Western blots showing serial changes in endogenous DUSP6 protein levels in 293 T cells stably expressing an empty vector or HA-KRAS$^{G12V}$ (**b**) or in Pa01c cells stably expressing a scramble sequence or an shRNA against *KRAS* (**c**), after treatment with 10 µg/mL cycloheximide (CHX) for the indicated durations. **d** Western blots, and quantification of serial endogenous DUSP6 protein levels in

Pa01c cells stably expressing a scramble sequence or an shRNA against *KRAS* (**c**), after treatment with 2 µM of ulixertinib for the indicated durations. **e** Western blots, and quantification of serial endogenous DUSP6 in Pa01c cells pre-treated with DMSO or MRTX1133 (0.5 µM) for 1 h followed by 2 µM of ulixertinib at the indicated durations. **f** Western blots, and quantification of serial endogenous DUSP6 in MIA Paca-2 cells pre-treated with DMSO or AMG-510 (0.5 µM) for 1 h, followed by 2 µM of ulixertinib for the indicated durations. For (**b–f**), half-lives ($t_{1/2}$) of DUSP6 were calculated by measuring the DUSP6 band intensities, normalizing to $t_0$ and performing one-phase exponential decay analysis, as shown in the graph below. **a** was conducted three times, and (**b–f**) were conducted two times, and one set of data for each was shown. Source data are provided in Source Data file.

---

more and larger colonies, but this effect was blocked by afatinib (Fig. 5g), supporting a role of HER2 signaling in driving this phenotype. As separate unbiased confirmation, TRIM21-silenced HPAC cells displayed enhanced ERBB2 and KRAS signatures by RNAseq (normalized enrichment scores 6.52 and 3.35, respectively, false discovery rate $q = 0$ for both, Fig. 5h). Collectively, our data thus far depict a model in which MAPK inhibitors dissociate TRIM21 from DUSP6, causing DUSP6 to be proteasomally degraded, subsequently leaving HER2 in a more phospho-activated state which contributes to further DUSP6 degradation while simultaneously drives other survival pathways such as the PI3K-AKT cascade (Fig. 5i).

**Targeted MAPK-based combinations require cytotoxic chemotherapy to achieve meaningful therapeutic efficacy**

Thus far, our data have depicted a mechanistic link between MAPK inhibition and HER2 activation, prompting us to test the combination of MAPK inhibitors plus afatinib in PDX models. We utilized ulixertinib, a 1st-in-class ERK inhibitor that has shown single-agent activity in NRAS-mutant melanoma patients[15]. Because MAPK inhibition also activates the pro-survival PI3K-AKT cascade (Fig. 1i), as also widely reported by others[16,45–47], we included the combination of ulixertinib

and the PI3K inhibitor copanlisib for comparison of efficacy. In a pilot experiment using six KRAS-mutant PDX models, although ulixertinib plus afatinib or copanlisib were more effective than single agent alone in curbing tumour growth (Fig. 6a), no tumour regression was observed. Because gemcitabine can synergize with ulixertinib (Supplementary Fig. 1f), we added gemcitabine to these two doublet regimens and performed an efficacy analysis in 16 early passaged PDXs, fourteen of which had a KRAS mutation[6] (Fig. 6b). We found that adding gemcitabine to either ulixertinib + copanlisib or ulixertinib + afatinib was more effective in inhibiting PDX tumour growth than the ulixertinib + gemcitabine doublet (Supplementary Fig. 5a). Using the clinical RECIST 1.1 criteria[48], the partial response (PR), stable disease (SD), and disease control rates (PR + SD, DCR) for each treatment group after 4 weeks of treatment were gemcitabine:0%, 6%, and 6%; gemcitabine + ulixertinib:6%, 0%, and 6%; gemcitabine + ulixertinib + afatinib:12.5%, 25%, and 37.5%; gemcitabine + ulixertinib + copanlisib: 37.5%, 31.3%, and 68.5%, respectively (Fig. 6c). Although the gemcitabine + ulixertinib + copanlisib triplet appeared superior to other combinations, the treated mice experienced significant weight loss after 2–4 weeks of treatment (Fig. 6d, Supplementary Fig. 5b), leading to mandatory treatment breaks. After the treatment break, some mice

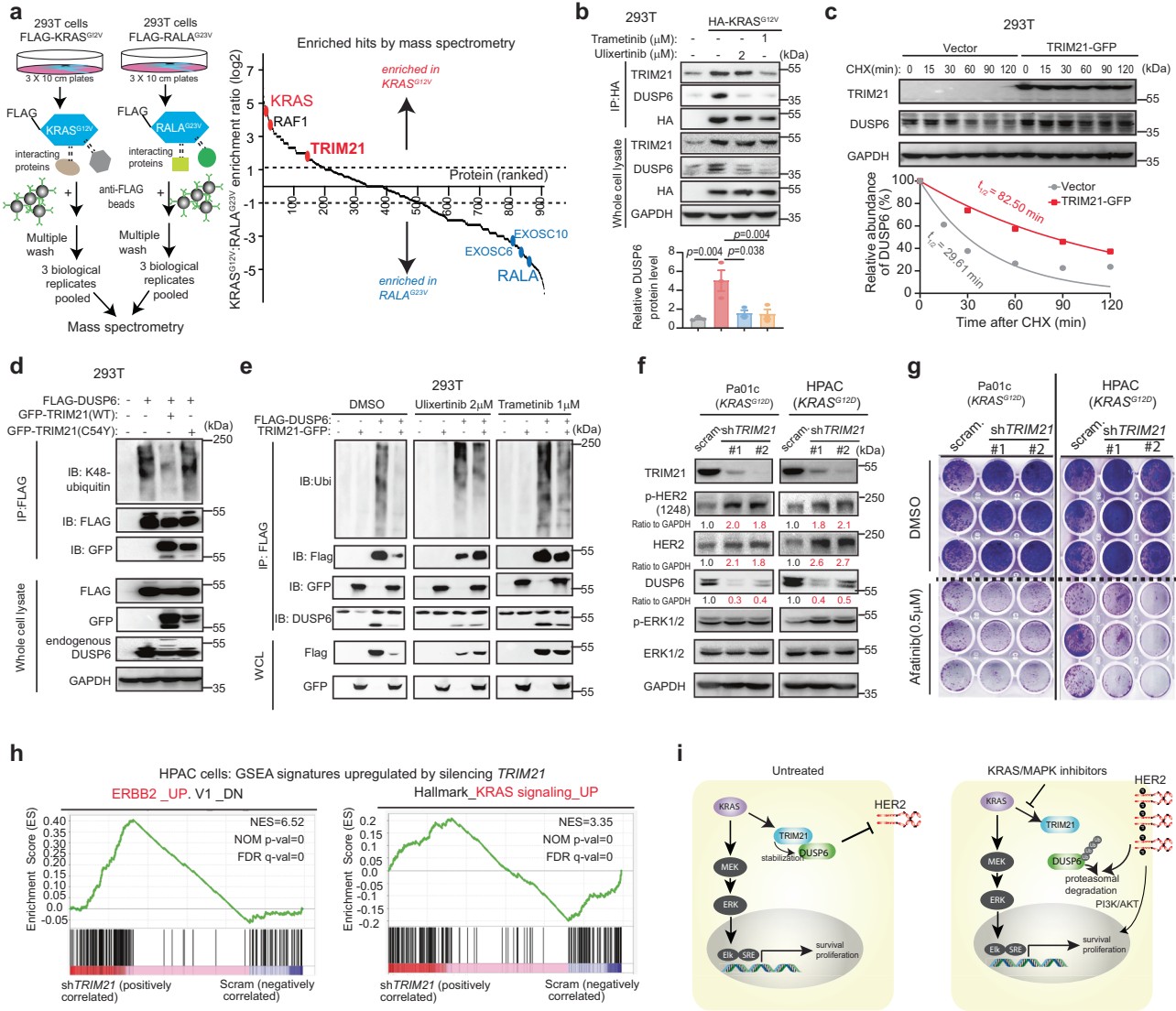

**Fig. 5 | KRAS recruits TRIM21 to regulate DUSP6 stability. a** Schematic of the immunoprecipitation followed by mass spectrometry (IP-MS) experiment employed to identify binding partners of FLAG-KRAS$^{G12V}$. 3 biological replicates from each condition were pooled for mass spectrometry. Linear plot shows the enriched proteins. **b** IP western blots showing the interaction between HA-KRAS$^{G12V}$ and endogenous TRIM21 and DUSP6 in 293 T cells stably expressing vector or HA-KRAS$^{G12V}$ treated with DMSO, trametinib, or ulixertinib for 16 h. The bar graph includes quantification of three independent experiments. Data was presented as mean ± SEM. *P*-values were calculated using one-way ANOVA followed by Dunnet's multiple comparisons test. **c** Western blots, and quantification of serial endogenous DUSP6 protein levels in 293 T cells expressing empty vector or GFP-tagged TRIM21 following treatment with CHX (10 μg/mL) for the indicated durations. **d** IP experiment showing differences in K48-polyubiquitination of FLAG-tagged DUSP6 in 293 T cells transfected with the vector, wild-type, or enzymatically inactive (C54Y) TRIM21. **e** IP experiment showing differences in K48-polyubiquitination of FLAG-

tagged DUSP6 with/without TRIM21 co-expression following 16 h treatment with DMSO, ulixertinib, or trametinib in 293 T cells. **f** Western blots showing changes in p-HER2, p-ERK1/2, and DUSP6 in Pa01c and HPAC cells stably expressing a scramble shRNA or two different shRNAs against *TRIM21*. **b** was conducted three times, (**c**–**f**) were conducted two times, and one set of data for each was shown. **g** 2-dimensional colony formation assay showing the colony-forming ability of the indicated Pa01c and HPAC cells co-treated with DMSO or afatinib for 2 weeks. **h** GSEA plots showing the enrichment of ERBB2 and KRAS signatures in *TRIM21*-silenced HPAC cells subjected to bulk RNA sequencing. **i** proposed model of the mechanism by which KRAS/MAPK inhibitors dissociate TRIM21 from DUSP6 and causes the latter to be destabilized, leading to sustained phospho-activation of HER2. Phospho-activated HER2 may in turn promotes DUSP6 degradation while simultaneously activate other survival cascades such as the PI3K-AKT pathway. Source data are provided in Source Data file.

failed to recover and had to be euthanized (WU-0007, WU-0011, WU-0022). For the remaining 13 PDX models, where the mice had recovered enough to resume treatment, only two PDX models (WU-0009 and WU-0016) showed disease progression, whereas the remaining 11 treated PDX models had stable disease. Many of these re-treated mice experienced weight loss again and later had to be euthanized despite having small tumours. In contrast, the gemcitabine + ulixertinib + afatinib triplet appeared to be better tolerated as the treated mice did not exhibit significant weight loss during treatment (Supplementary Fig. 5a, b). These experiments show that co-targeting the MAPK and

HER signaling with inhibitors may be inadequate and require an additional cytotoxic agent to achieve meaningful anti-tumour efficacy in PDAC.

### MAPK inhibitors plus antibody-drug conjugate trastuzumab deruxtecan (DS-8201a) lead to deep and durable treatment response

Although the gemcitabine + ulixertinib + afatinib triplets showed more efficacy in PDX models, the DCR was merely 37.5%. In addition, afatinib and ulixertinib share overlapping side effects, including skin rash,

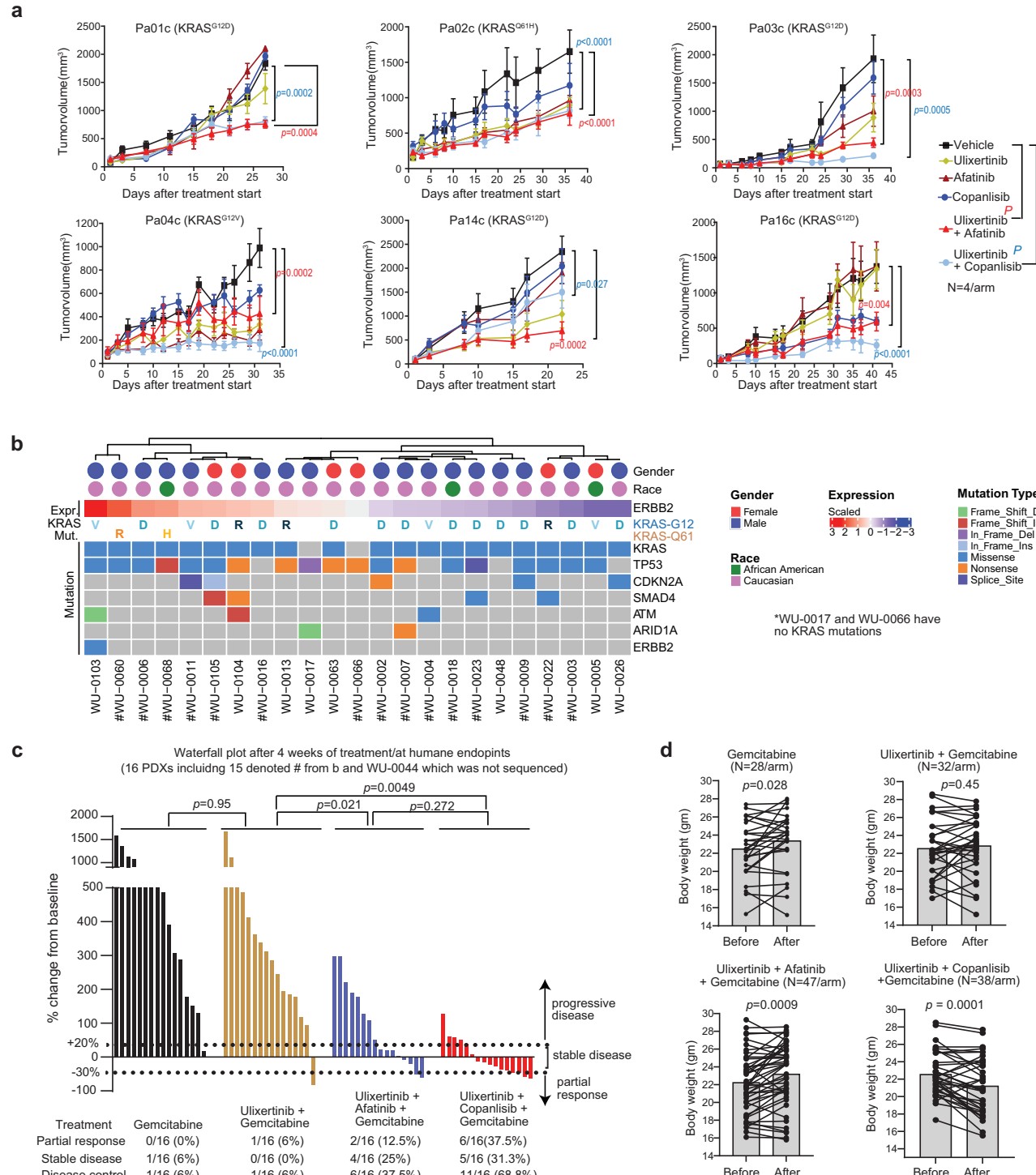

**Fig. 6 | Targeted MAPK-based combinations require cytotoxic chemotherapy to achieve meaningful therapeutic efficacy. a** Growth kinetics of the indicated PDCL subcutaneous xenograft tumours treated as indicated when the tumours reached 50–100 mm³ in NOD-SCIDγ mice. Dosages of each agent were: ulixertinib 100 mg/kg BID orally 5 days/week, afatinib 12.5 mg/kg orally daily, copanlisib 10 mg/kg by tail vein injection two times/week, gemcitabine 75 mg/kg by intraperitoneal injection once per week. *N* = 4 tumours per treatment. Data was presented as mean ± SEM. *P*-values were calculated using one-way ANOVA with Dunnet's multiple comparisons test. **b** Clinical and genomic profiles and *ERBB2* mRNA expression in PDAC PDX models used in this study. **c** Waterfall plot summarizing changes of tumour volume from baseline for all 16 PDX models and determination of treatment response using clinical RECIST 1.1. *P*-values were calculated using Brown-Forsythe and Welch ANOVA test followed by Dunnett's T3 multiple comparisons test. **d** Paired data plots showing changes in the body weight of mice treated with different regimens for 4 weeks. *P*-values were calculated by two-tailed paired *t*-test. Loss in body weight after two weeks of treatment as indicated. All data are provided in the Source Data file.

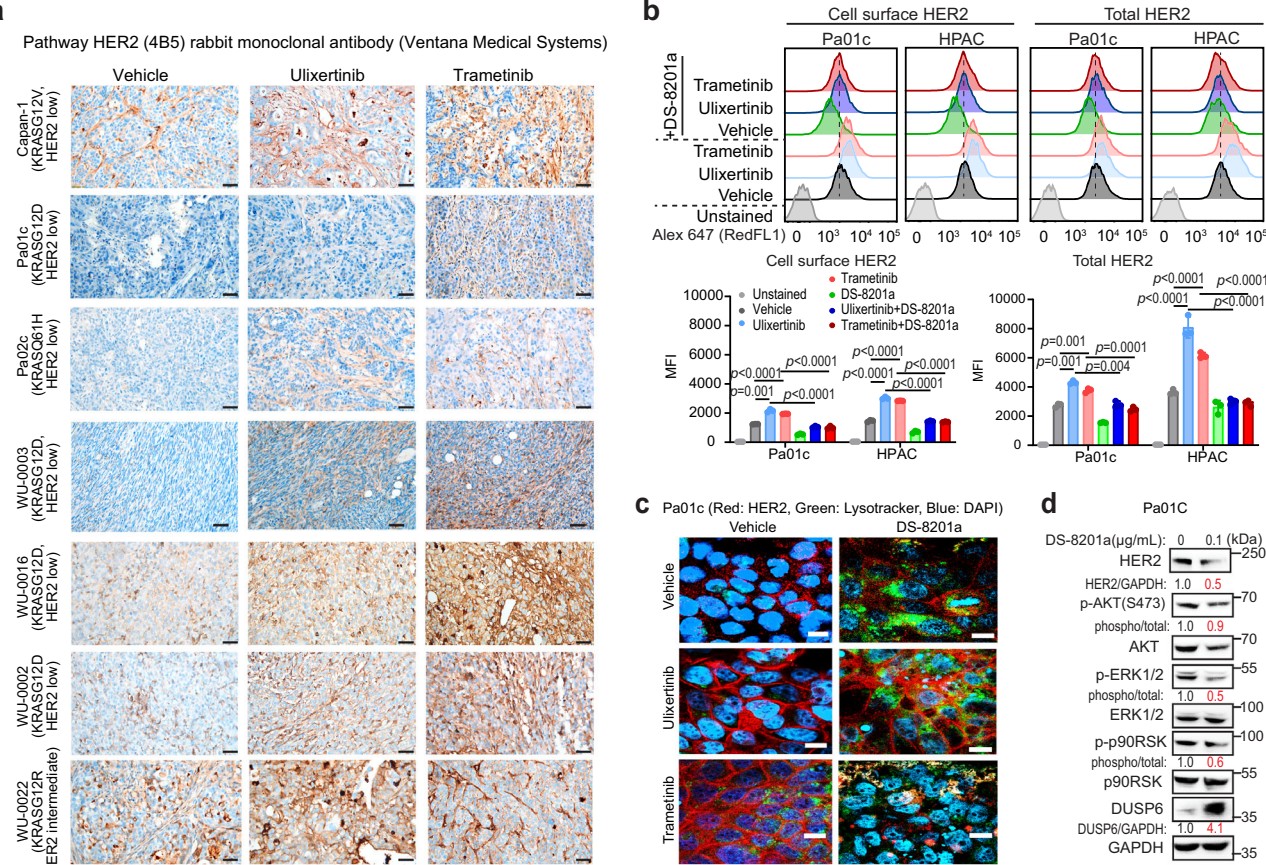

**Fig. 7 | MAPK inhibitors upregulate HER2 expression. a** Representative IHC images showing changes in HER2 protein expression in the indicated PDAC xenografts using a CLIA-certified HER2 staining kit routinely used for the analysis of gastric cancer samples, Scale bars = 20 μM. **b** Representative FACS plots and quantification showing changes in surface (without cell permeabilization) and total (with cell permeabilization) HER2 abundance following 16 h treatment as indicated (ulixertinib 2 μM, trametinib 0.5 μM, DS-8201a 0.1 μg/ml). Data represents one of three independent experiments each done in triplicates. Data are presented as the mean ± SEM. *P*-values were calculated using unpaired *t*-test with Welch's correction or ANOVA followed by Tukey's test. **c** Representative IF images showing increased surface and total HER2 expression (red) in Pa01c cells treated for 16 h with the indicated agents as (**b**). LysoTracker Green DND-26 was used to stain the endolysosomes, Scale bars = 50 μM. **d** Western blots showing suppression of MAPK activity and upregulation of DUSP6 in Pa01c cells after treatment with vehicle or DS-8201a 0.1 μg/ml for 16 h. **d** was conducted two times, and one set of data was presented. All data are provided in Source Data file.

fatigue, diarrhea, and anorexia[5], and thus may not be well tolerated in patients. Because ulixertinib and trametinib upregulate HER2 expression, this provided us with an opportunity to test trastuzumab deruxtecan (DS-8201a), a humanized monoclonal anti-HER2 antibody (trastuzumab) conjugated with a topoisomerase 1 inhibitor (deruxtecan). Once bound to HER2, DS-8201a is internalized into the endolysosomes and cleaved by lysosomal enzymes, including cathepsins B and L, which are highly expressed in tumour cells[49–51], where deruxtecan is released. Using a clinically approved HER2 IHC staining protocol, we observed increased HER2 expression in xenograft tumours treated with either ulixertinib or trametinib (Fig. 7a). By flow cytometry, both ulixertinib and trametinib significantly increased surface and total HER2 expression in Pa01C and HPAC cells, which was reduced by DS-8201a (Fig. 7b). Fluorescence microscopy showed that trametinib and ulixertinib upregulated HER2 expression at the plasma membrane, and co-treatment with DS-8201a resulted in internalization of HER2 into the endolysosomes, as labelled using LysoTracker Green (Fig. 7c). Western blotting showed that DS-8201a induced down-regulation of total HER2 and p-ERK1/2 and upregulation of DUSP6 (Fig. 7d), supporting our model that HER2 and DUSP6 negatively regulate each other.

Next, we tested the in vivo antitumour activity of MEK or ERK inhibitors in combination with DS-8201a in PDAC xenografts. In all nine models tested, DS-8201a or MAPK inhibitors alone showed modest suppression of tumour growth. Strikingly, MEK or ERK inhibitors combined with DS-8201a resulted in sustained tumour regression in eight out of nine PDX models (Fig. 8a). In the WU-0022 model, which harbors a KRAS[G12R] mutation, tumours recurred after treatment discontinuation, but regression was again achieved after re-treatment. By the RECIST 1.1 criteria, the ulixertinib + DS-8201a or trametinib + DS-8201a doublets achieved complete or near-complete response in all tested models except for WU-0009, which continued to grow at a lower kinetic (Fig. 7b). In contrast, trastuzumab plus trametinib slowed but did not regress tumour growth (Fig. 7c), again signifying the need for a cytotoxic agent for this strategy. Notably, mice treated with trametinib or ulixertinib plus DS-8201a retained their body weight throughout treatment and during observation after tumour regression (Supplementary Fig. 6a). Furthermore, mice treated with trametinib + DS-8201a showed no significant abnormalities in blood chemistries compared to those treated with the vehicle of a single agent (Supplementary Fig. 6b). At the time of euthanasia, we did not observe histological abnormalities in the lungs, liver, kidneys, and intestines of combo-treated mice (Supplementary Fig. 6c).

Because each MEK inhibitor has a different mechanism of action, we tested whether other MEK inhibitors can similarly upregulate total or phospho-HER2 expression in PDAC cells. We found that mirdametinib and selumetinib also increased phospho-HER2

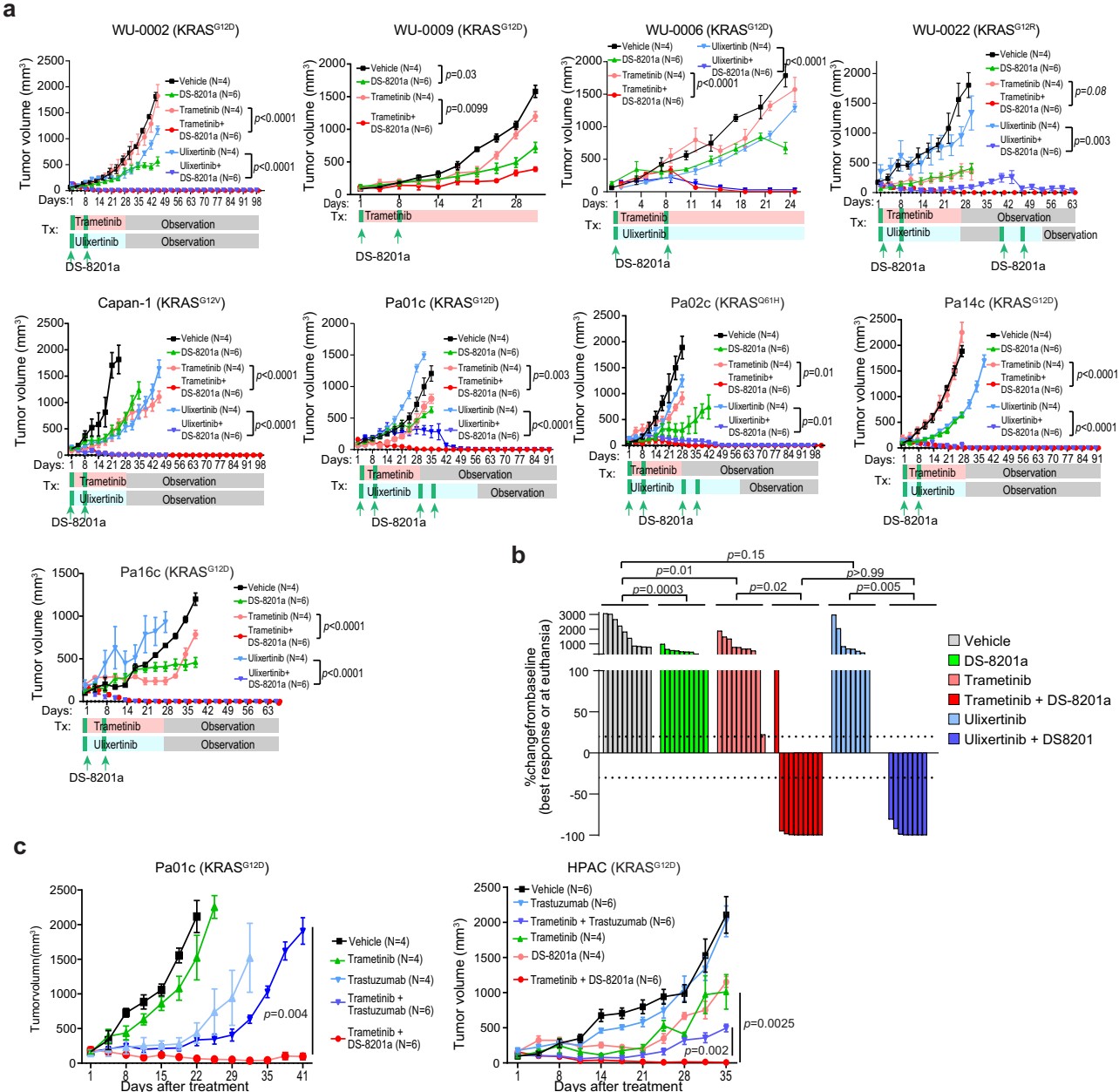

**Fig. 8 | MAPK inhibitors plus antibody-drug conjugate trastuzumab deruxtecan (DS-8201a) lead to deep and durable treatment response. a** Growth kinetics of the subcutaneous PDAC tumours treated as indicated when tumours reached ~100 mm³ in NOD-SCIDγ mice. WU0002, WU0006, WU0022 Capan-1, Pa02c, Pa14c & Pa16c (DS-8201a + ulixertinib vs ulixertinib or DS-8201a + trametinib vs trametinib: $p < 0.0001$), for WU0009 (trametinib vs vehicle: $p = 0.9998$, DS-8201a vs vehicle: $p = 0.0043$ & DS-8201a + trametinib vs trametinib: $p = 0.0043$), for Pa01c (DS-8201a + ulixertinib vs ulixertinib: $p = 0.0027$ & DS-8201a + trametinib vs trametinib ulixertinib: $p < 0.0001$). **b** Waterfall plot summarizing changes of

tumour volume for different PDAC models treated as indicated. Comparison to the baseline was made using the best response for combo-treated mice or when mice were euthanized. The treatment response was determined using the clinical RECIST 1.1 criteria. Each bar represents the average change in volume from replicates of a different model. *P*-values calculated using one-way ANOVA with Tukey's multiple comparison test. **c** Growth kinetics of the subcutaneous PDAC models treated as indicated when tumours reached between 50–100 mm³ or above in NOD-SCIDg mice. **a**–**c** data are presented as the mean ± SEM. *P*-values were calculated using ANOVA followed by Tukey's test. Source data are provided in the Source Data file.

levels by Western blots (Supplementary Fig. 7a), as well as total and surface HER2 expression in Pa01c and HPAC cells by flow cytometry (Supplementary Fig. 7b). Together, these data provide a compelling rationale for evaluating MEK or ERK inhibitors plus DS-8201a in PDAC clinical trials.

### KRAS inhibitors plus antibody-drug conjugate trastuzumab deruxtecan (DS-8201a) showed promising preclinical efficacy

The development of KRAS inhibitors has been one of the biggest therapeutic breakthroughs in recent years. Because KRAS inhibitors

AMG-510 and MRTX1133 downregulate DUSP6 (Fig. 4e, f), which has also been observed by others[52–54], we hypothesized that these inhibitors can similarly upregulate HER2 expression. Using FACS assay, we found that AMG-510 upregulated surface HER2 expression in different KRAS[G12C]-mutant cell lines, including lung adenocarcinoma NCI-H2122 and NCI-H2030 (Fig. 9a) and colon adenocarcinoma SW837 and SW1463 (Supplementary Fig. 8a). The combination of AMG-510 and DS-8201a showed strong synergism in curbing the growth of all three cell lines in vitro, with most CI values falling below 0.1 (Fig. 9b). This combination led to a partial response

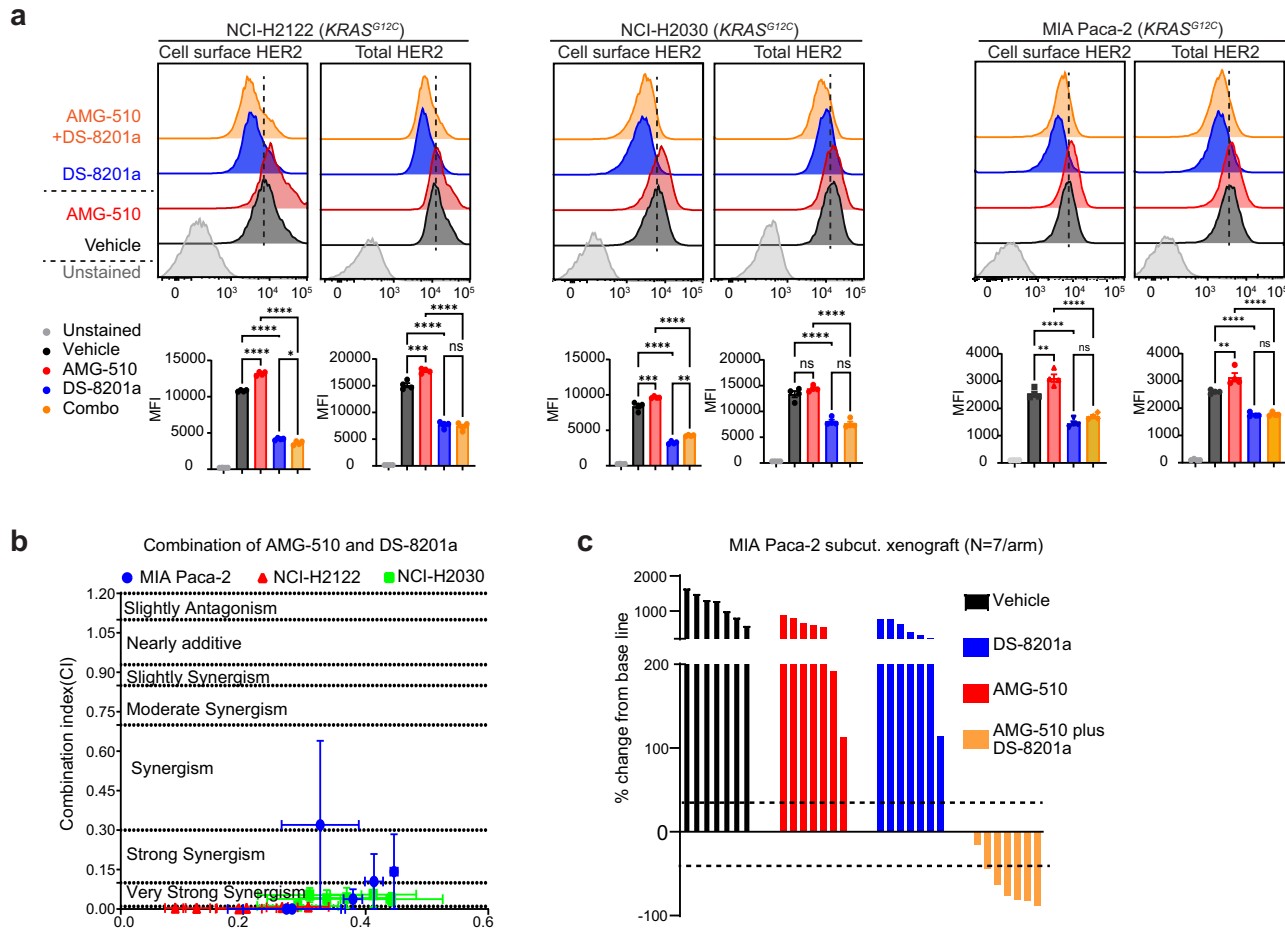

**Fig. 9 | KRASG12C inhibitor plus trastuzumab deruxtecan showed promising preclinical efficacy. a** Representative FACS plots and quantification showing changes in surface and total HER2 abundance following 16 h treatment, as indicated in three different *KRAS^G12C*-mutant cell lines. (AMG-510 0.5 μM, DS-8201a 0.1 μg/ml). Data represents one of three independent experiments each done in triplicates. Data are presented as the mean ± SEM. *P*-values were calculated using one way ANOVA followed by Tukey's multiple comparisons test. NCL-H2122 Surface HER2 (AMG-510 vs vehicle: *p* < 0.0001, DS-8201a vs vehicle: *p* < 0.0001, DS-8201a + AMG-510 vs AMG-510: *p* < 0.0001 & DS-8201a + AMG-510 vs DS-8201a: *p* = 0.0120), and Total HER2 (AMG-510 vs vehicle: *p* = 0004, DS-8201a vs vehicle: *p* < 0.0001, DS-8201a + AMG-510 vs AMG-510: *p* < 0.0001 and DS-8201a + AMG-510 vs DS-8201a: *p* = 0.8964); NCL-H2030 Surface HER2 (AMG-510 vs vehicle: *p* = 0.0006, DS-8201a vs vehicle: *p* < 0.0001, DS-8201a + AMG-510 vs AMG-510: *p* < 0.0001 & DS-8201a + AMG-510 vs DS-8201a: *p* = 0041), and Total HER2 (AMG-510 vs vehicle: *p* = 0.3401, DS-8201a vs vehicle: *p* < 0.0001, DS-8201a + AMG-510 vs AMG-510: *p* < 0.0001 and DS-8201a + AMG-510 vs DS-8201a: *p* = 0.9033; MIA Paca-2 Surface HER2 (AMG-510 vs vehicle: *p* = 0.0047, DS-8201a vs vehicle: *p* < 0.0001, DS-8201a + AMG-510 vs

AMG-510: *p* < 0.0001 & DS-8201a + AMG-510 vs DS-8201a: *p* = 0.2871), and Total HER2 (AMG-510 vs vehicle: *p* = 0.0030, DS-8201a vs vehicle: *p* < 0.0001, DS-8201a + AMG-510 vs AMG-510: *p* < 0.0001 and DS-8201a + AMG-510 vs DS-8201a: *p* = 0.9985). **b** Median effect analyses of AMG-510 in combination with DS-8201a in three *KRAS^G12C*-mutant cell lines, as represented by combination indices (CI) calculated using Compusyn software. Cells were cultured in triplicate at six fixed-ratio concentrations (1:1, 0.5:0.5, 0.25:0.25, 0.125:0.125, 0.063:0.063, and 0.031:0.031) for 3 days, and viability was measured using the Alamar Blue assay. **c** Waterfall plot summarizing changes in tumour volume for each MIA Paca-2 tumour. All mice were euthanized simultaneously when vehicle-treated mice reached the humane endpoints, and comparison to the baseline was made. The treatment response was determined using the clinical RECIST 1.1 criteria. Each bar represents an individual tumour. *N* = 7/arm. *P*-values were calculated using one-way ANOVA followed by Tukey's multiple comparison test. AMG-510 vs vehicle (*p* = 0.0007), DS-8201a vs vehicle (*p* = 0.0002), DS-8201a + AMG-510 vs vehicle (*p* < 0.0001), DS-8201a + AMG-510 vs DS-8201a (*p* = 0084) and DS-8201a + AMG-510 vs DS-8201a (*p* = 0022).

in seven out of eight MIA Paca-2 tumours grown in immunocompromised mice, while neither agent alone showed meaningful efficacy compared to the vehicle (Fig. 9c). Similarly, MRTX1133 upregulated the surface expression of HER2 in KRAS^G12D-mutant Pa01c and HPAC cells, as determined by immunofluorescence (Fig. 10a) and flow cytometry (Fig. 10b). Knockdown of ERBB2, and to a much lesser extent ERBB1 or ERBB3, led to increased sensitivity of Pa01c and HPAC cells to MRTX1133 in vitro (Supplementary Fig. 8b). The combination of DS-8201a and MRTX1133 at 10 mg/kg, which has little single-agent efficacy, led to partial response in four out of eight Pa01c xenograft tumours (Fig. 10c). These data support testing the combination of KRAS inhibitors and DS-8201a in future clinical trials.

## Discussion

As of now, combination chemotherapy remains the primary treatment approach for all PDAC patients. Targeted therapies and immunotherapeutics are effective only for a small subset of patients, typically those with specific genetic alterations, deficient mismatch repair or high tumour mutational burden. Our study underscores the critical role of the MAPK pathway in PDAC, not solely driven by the KRAS oncoprotein but also by therapeutic stress. We provided mechanistic evidence showing that MAPK and KRAS inhibitors play a role in destabilizing the DUSP6 protein, consequently leading to the activation of HER2. This adaptive mechanism can be exploited by incorporating a HER2-directed antibody-drug conjugate (ADC), resulting in sustained tumour regression in PDAC PDXs.

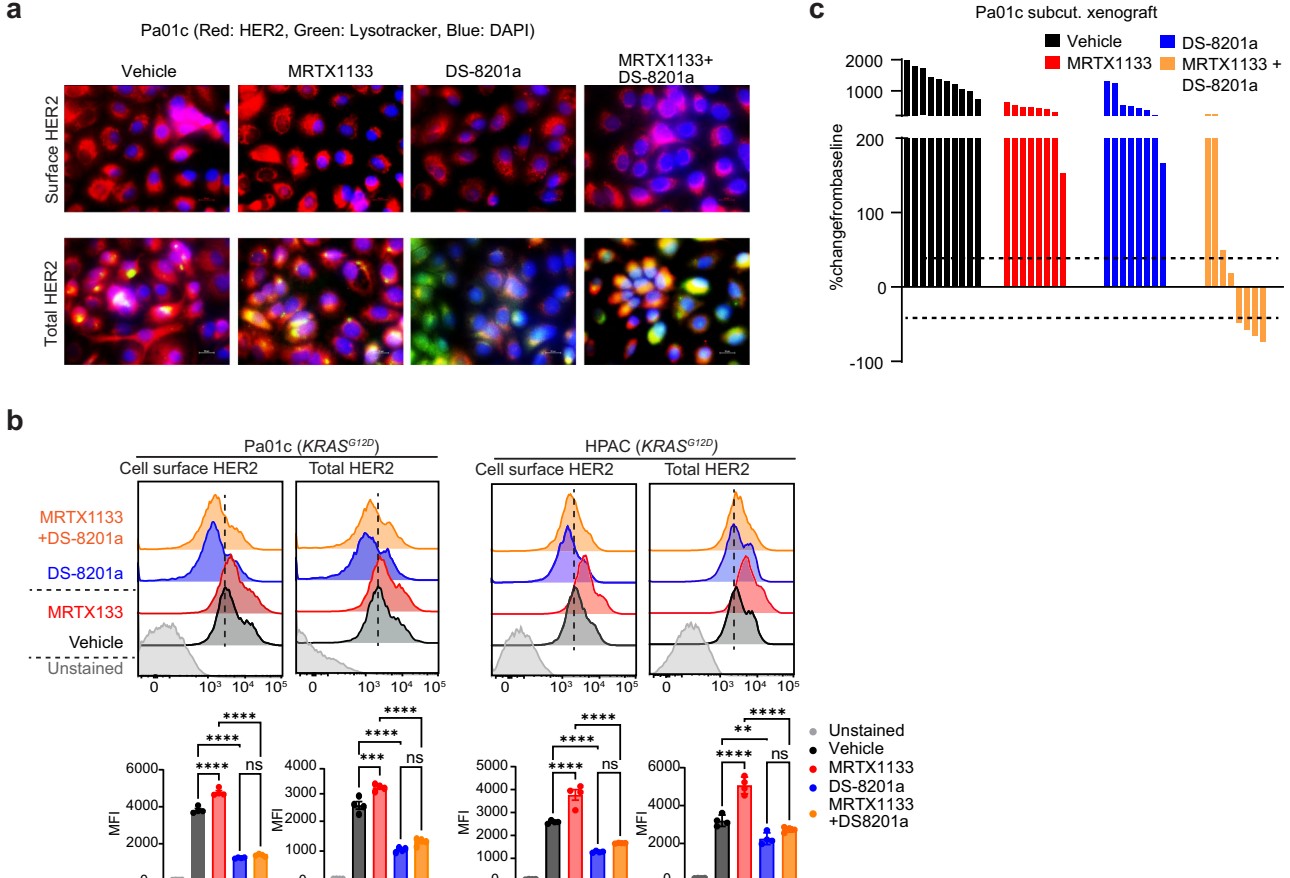

**Fig. 10 | KRASG12D inhibitor plus trastuzumab deruxtecan showed promising preclinical efficacy. a** Representative IF images showing increased surface and total HER2 expression (red) in Pa01c cells treated for 16 h with the indicated agents. LysoTracker Green DND-26 was used to stain the endolysosomes. (MRTX1133 0.5 μM, DS-8201a 0.1 μg/ml). Scale bars = 20 μM. **b** Representative FACS plots and quantification showing changes in surface (without cell permeabilization) and total (with cell permeabilization) HER2 abundance following 16 h treatment, as indicated in two different *KRAS^{G12D}*-mutant cell lines. (MRTX1133 0.5 μM, DS-8201a 0.1 μg/ml). Data represents one of three independent experiments each done in triplicates. Data are presented as the mean ± SEM. *P*-values were calculated using one way ANOVA followed by Tukey's multiple comparisons test. Pa01c Surface HER2: MRTX1133 vs vehicle ($p < 0.0001$), DS-8201a vs vehicle ($p < 0.0001$), DS-8201a + MRTX1133 vs MRTX1133 ($p < 0.0001$) & DS-8201a + MRTX1133 vs DS-8201a ($p = 0.1598$), and Total HER2: MRTX1133 vs vehicle ($p = 0.0008$), DS-8201a vs vehicle ($p < 0.0001$), DS-8201a + MRTX1133 vs MRTX1133 ($p < 0.0001$) and DS-

8201a + MRTX1133 vs DS-8201a ($p = 0.1215$). HPAC Surface HER2: MRTX1133 vs vehicle ($p < 0.0001$), DS-8201a vs vehicle ($p < 0.0001$), DS-8201a + MRTX1133 vs MRTX1133 ($p < 0.0001$) & DS-8201a + MRTX1133 vs DS-8201a ($p = 0.1502$), and Total HER2: MRTX1133 vs vehicle ($p < 0.0001$), DS-8201a vs vehicle ($p = 0.0055$), DS-8201a + MRTX1133 vs MRTX1133 ($p < 0.0001$) and DS-8201a + MRTX1133 vs DS-8201a ($p = 0.1926$). **c** Waterfall plot summarizing changes in tumour volume for each Pa01c tumour, as indicated. All mice were euthanized simultaneously when vehicle-treated mice reached the humane endpoints, and comparison to the baseline was made. The treatment response was determined using the clinical RECIST 1.1 criteria. Each bar represents an individual tumour. $N = 7$. *P*-values were calculated using one-way ANOVA followed by Tukey's multiple comparison test. MRTX1133 vs vehicle ($p < 0.0001$), DS-8201a vs vehicle ($p = 0.0001$), DS-8201a + MRTX1133 vs vehicle ($p < 0.0001$), DS-8201a + MRTX1133 vs MRTX1133 ($p = 0.0867$) and DS-8201a + MRTX1133 vs DS-8201a ($p = 0083$). Source data are provided in Source Data file.

Co-targeting of KRAS-MAPK and adaptive EGFR/HER1 signaling is a well-proven strategy in colon cancer. In BRAF^{V600E} colon cancer, the combination of encorafenib, binimetinib, and cetuximab resulted in an ORR of 26% and a median overall survival of 9 months[55]. In heavily pre-treated KRAS^{G12C}-mutant colon cancer patients, adagrasib plus cetuximab produced an impressive 46% response rate[56]. Our PDX data showed that the combination of ulixertinib and afatinib did not lead to tumour regression and required the addition of a cytotoxic agent. The advent of DS-8201a (trastuzuman-deruxtecan) provides an attractive solution to fulfill the goals of curbing MAPK and HER2 signaling, while including guided delivery of a cytotoxic. DS-8201a monotherapy significantly prolonged the survival of patients with HER2-positive gastric[57], breast[58], colon cancers[59] and HER2-mutant non-small cell lung cancer[60]. More excitingly, DS-8201a demonstrated a response rate of ~50% in HER2-low (IHC1 + and 2+) metastatic breast cancer patients, leading to FDA approval for this population[61]. HER2 IHC 2 + or 3 + staining has been reported in 16–61% of PDAC samples[30,62,63]. While

this wide variation is due to a lack of standardized staining methodology, it can be safely assumed that a significant portion of PDAC samples display some level (1 + to 3+) of HER2 expression, which could potentially be further augmented by MAPK or KRAS inhibitors to render DS-8201a more effective. Equally importantly, because DS-8201a does not recognize murine HER2, the lack of systemic toxicities observed in mice treated with DS-8201a and trametinib must not be overinterpreted, and proper toxicity and pharmacokinetic assessments are warranted in future clinical trials.

The recent advent of KRAS^{G12C} inhibitors (KRASi), including sotorasib and adagrasib, is a major therapeutic breakthrough for lung adenocarcinoma, in which KRAS^{G12C} mutation is more common[64–66]. In PDAC, where KRAS^{G12C} mutation constitutes ~1% of all cases, sotorasib showed an ORR of ~20% and progression-free survival of ~4 months[67]. Although KRAS^{G12D} inhibitors are still being tested in early phase clinical trials, similar efficacy to sotorasib is anticipated. Here, we showed that the combination of KRAS inhibitors and DS-8201a is highly

promising. Because KRAS[G12C] and KRAS[G12D] inhibitors are expected to exclusively target and upregulate HER2 in KRAS-mutant cells, the systemic toxicity of this combination should be limited.

We provide experimental evidence that DUSP6 is a phosphatase that binds and dephosphorylates HER2, at least in part mediated by the C-terminal TEY motif of HER2. We showed that destabilization of DUSP6 following MAPK inhibition resulted in sustained HER2 phosphorylation. Prior to this study, two other phosphatases have been shown to dephosphorylate HER2. Protein tyrosine phosphatase 18 (PTPN18 or BDP1) is the first phosphatase to inhibit HER2 phosphorylation and MAPK activity[68]. Using an RNAi screen, PTPN13 was also found to suppress HER2 phosphorylation and MAPK and PI3K signaling[69]. In the presence of KRAS, DUSP6 is stabilized by TRIM21 via its interaction with KRAS. This allows DUSP6 to negatively regulate ERK1/2 and inhibit mitogenic signaling, which is toxic to KRAS-mutant cells[70]. In the presence of MAPK inhibitors, TRIM21 dissociates from DUSP6, causing it to be degraded and resulting in the sustained phosphorylation of HER2. Nonetheless, much of the molecular details of these events remain to be fully understood. First, it remains unclear how MAPK inhibitors dissociate TRIM21 from DUSP6. One possibility is that inhibitor bound MEK and ERK molecules are tightly bound to DUSP6 and preclude it from interacting with TRIM21. Second, the bona fide E3 ligase that polyubiquitinates and degrades DUSP6 remains elusive in this context. FBXO31 has been shown to polyubiquitinate and degrade DUSP6[71]; thus, it would be interesting to determine whether FBXO31 is involved in MAPK/KRAS inhibitor induced DUSP6 degradation. Third, it is unclear how TRIM21 stabilizes DUSP6. It will be interesting to determine whether TRIM21 competes with FBXO31 or other E3 ligases for DUSP6 and protects it from degradation.

In summary, our study offers valuable insights into the potential therapeutic strategies for PDAC by targeting the KRAS-MAPK and HER2 pathways. The promising combination of KRAS/MAPK inhibitors and an anti-HER2 ADC warrants further investigation in clinical trials for PDAC patients, and potentially other KRAS-mutant cancer types.

## Methods

### Study approval and rigors
All procedures performed in this study involving animals were in accordance with the ethical standards of the institutional and/or national research committee and with the 1964 Helsinki Declaration and its later amendments or comparable ethical standards and were approved by Washington University IACUC (Protocol #22-0101). Animal experiments and analyses were conducted in a blinded manner by independent laboratory members to ensure rigor. In vitro experiments were replicated two–four times and with different cell lines.

### Cell lines
All cell lines, including HPAC, MIA Paca-2, Capan-1, NCI-H2122, NCI-H2030, SW837, SW1463 and 293 T cells, were purchased from ATCC and authenticated on their own cell lines. Pa01c, Pa02c, Pa03c, Pa14c, and Pa16c were kind gifts from Dr. Channing Der at UNC-CH and whole-exome sequenced[72]. All lines were used for <6 months after receipt or resuscitation from cryopreservation. All cell lines were cultured in DMEM supplemented with 10% fetal bovine serum and 1% penicillin/streptomycin. Mycoplasma testing was performed semi-annually using a MycoSEQ Detection kit (Applied Biosystems).

### Reverse phase protein array (RPPA)
Pa01c and HPAC cell lysates were prepared using pre-made lysis buffer provided by the RPPA core at MD Anderson Cancer Center. Samples were probed with antibodies by tyramide-based signal amplification approach and visualized by DAB colorimetric reaction. Slides were scanned on a flatbed scanner to produce 16-bit tif image. Spots from tif images were identified and the density was quantified by Array-Pro Analyzer. All the data points were normalized for protein loading and

transformed to linear value, designated as "Normalized Linear." "Normalized Linear" values were transformed to Log2 values, and median-centered for analysis.

### Immunoblotting
Western blotting was performed according to standard procedures. After appropriate treatment, the medium was removed, and cells were washed twice with ice-cold 1X PBS and lysed with ice-cold 1% Triton-X lysis buffer (25 mM Tris, pH 7.4, 150 mM NaCl, 5 mM EDTA, 1% Triton-X) containing 1X protease (10 μg mL$^{-1}$ leupeptin, 700 ng mL$^{-1}$ pepstatin, 170 ng mL$^{-1}$ aprotinin, 1 mM PMSF) and phosphatase (10 mM NaF, 1 mM Na$_3$VO$_4$, 1 mM Na$_4$P$_2$O$_7$, 5 mM Na β-glycerophosphate) inhibitors. lysate was transferred to a 1.5-ml microcentrifuge tube on ice and centrifuged at 13,500 rpm, 4 °C for 10 min. The pellet was discarded, and the supernatant was isolated, quantified by Bradford assay (Thermo Scientific), equalized for protein concentration, denatured, and reduced by 6X SDS sample buffer, and boiled for 5 min. 30–40 μg of proteins was resolved by SDS-PAGE and transferred to a Nitrocellulose membrane (Thermo scientific). Membranes were probed overnight at 4 °C with appropriate primary antibodies diluted in 5% bovine serum albumin (BSA), followed by appropriate HRP-conjugated secondary antibodies (anti-mouse or anti-rabbit, (1:2500 to 5000 dilution); Jackson Laboratory) and imaging using a chemiluminescent substrate (Pico or Femto, Thermo Fisher). Densitometry was performed using Image Lab software. Details on antibodies were provided in Supplementary Table 1.

### Protein degradation assay
293 T cells stably expressing HA-KRAS[G12V] and/or TRIM21-GFP were treated with cycloheximide (10 μg/ml) for the indicated times and immunoblotted to quantify the half-lives of endogenous DUSP6. The half-life was calculated using densitometry analysis of immunoblot images (ImageLab, Bio-Rad) and a one-phase exponential decay model (GraphPad Prism v8/9).

### Co-immunoprecipitation and immunoblot analysis
Stable cell lines expressing protein of interest or 293 T cells were plated on 10 cm plates and transiently transfected with different plasmids according to the experimental setting. Bortezomib was added for 6 h to allow visualization of polyubiquitination. Cells were washed with cold PBS twice and lysed with Triton-X lysis buffer (25 mM Tris, pH 7.4, 150 mM NaCl, 5 mM EDTA, 1% Triton-X) containing 1X protease and phosphatase inhibitors. Cells lysates were incubated with anti-HA (Thermo Scientific cat#88837) or Anti-FLAG M2 (Sigma-Aldrich cat#M8823) magnetic beads overnight at 4 °C, washed with TBS-T and TBS respectively, eluted in 2X SDS sample buffer, and boiled for 10 min at 95 °C, according to the manufacturer's protocol. Proteins were resolved by SDS-PAGE, blotted onto a Nitrocellulose membrane, and probed with primary antibodies (Supplementary Table 1).

### In vitro pulldown assays
Plasmid DNA constructs pET-29b(+) encoding human His6-tagged wild type C-terminus of HER2 (HER2C-term676-end), mutant (HER2C-term676-end TY/AA) and His6-tagged full length ERK2, wild type (ERK2 WT), mutant (ERK2TY/AA) were transformed into E. coli BL21(DE3) (Intact Genomics). The expression of His fusion proteins in BL21 cells was induced by adding 1 mM IPTG (GOLDBIO) to the bacterial culture medium (OD600) and incubating the cells for 4 h at 37 °C. Bacterial cells were harvested and sonicated in bacterial lysis buffer (GOLD-BIO#77-86-1) containing protease inhibitors. After removing the bacterial cell debris by centrifugation (12,000 g, 30 min), the supernatant was subjected to appropriate amount of Dynabeads™ His-Tag Isolation and Pulldown kit (Thermo Scientific cat#101034) in a total volume of 700 μL, 1X Binding/Wash Buffer (100 mM Sodium Phosphate, pH 8.0, 600 mM NaCl, 0.02% Tween™-20) and incubated overnight at 4 °C,

and then washed. To continue with pull-down assay, samples of interest prepared in pull-down buffer (6.5 mM Sodium phosphate, pH 7.4, 140 mM NaCl, 0.02% Tween™-20) in a total volume of 700 µL were added to the earlier bead-protein complex and incubate for 4 h at 4 °C. The beads were then washed four times with 1X Binding/Wash Buffer and eluted in His elution buffer (300 mM Imidazole, 50 mM Sodium phosphate pH 8.0, 300 mM NaCl, 0.01% Tween™-20) and boiled for 5 min in 1× sample buffer. Proteins were resolved by SDS-PAGE, blotted onto a Nitrocellulose membrane, and probed with primary antibodies (Supplementary Table 1).

### In vitro dephosphorylation and catalytic activation assay

In vitro dephosphorylation activity and catalytic activation of DUSP6 were performed using recombinant human DUSP6 protein (Abcam#ab183239). 293 T cells transfected with FLAG-HER2 was collected, lysed using TBS by freeze-thaw method and immunoprecipitated by Anti-FLAG M2 (Sigma-Aldrich cat#M8823) magnetic beads. The beads were then washed three times with ice cold TBS and immediately used. In vitro dephosphorylation of FLAG-HER2 was carried out by mixing 0.5 mg recombinant DUSP6 and Anti-FLAG M2 magnetic beads bound HER2 in 100 ml alkaline phosphatase stabilizing buffer (Sigma#A4955). The mixture was then incubated at 37 °C for 1 h on shaker. After the reaction, 2x SDS loading buffer was added into the mixture and heated at 100 °C for 10 min. 20 µl of the sample was used to detect the phosphorylation status of HER2 by Western blotting. In vitro phosphorylation activity and catalytic activation of DUSP6 was measured using general substrate p-Nitrophenyl Phosphate (PNPP, BioLabs Cat#P0757L). 2 mg recombinant DUSP6 was suspended in phosphatase buffer containing Anti-FLAG M2 magnetic bead-bound HER2 and p-NPP substrate in total volume of 100 µl. Following incubation at 37 °C for half an hour on shaker; then the released by product of p-NPP was measured at 405 nm.

### Duolink II proximity ligation assay (DPLA)

In situ interactions were detected by the Duolink II PLA kit per manufacturer's protocol (Sigma-Aldrich, St. Louis, MO, USA); PLA probe anti-rabbit Plus (Cat.#SLCL6875); PLA probe anti-mouse minus (5x, Cat.#SLCG6564); detection kit red (Cat.#Duo92008).

### Drugs

Drugs were obtained from the following: Gemcitabine and DS-8201a (Washington University Siteman Cancer Center Pharmacy), Paclitaxel (Selleckchem #S1150), 5-FU (Sigma #F6627), Oxaliplatin (Selleckchem #S1224), SN38 (Selleckchem #S4908), Hydroxychloroquine (Sigma #H0915), Trametinib (Selleckchem #S2673), Bortezomib (Selleckchem #S1013), Ruxolitinib(Selleckchem#s1378), MUC1 inhibitor (GO-201, Sigma#G7923), MMP inhibitor (R028-2653, AOBIOUS#AOB 2296), Mirdametinib (PD0325901, Selleckchem #S1036), Selumetinib (AZD6244, Selleckchem#S1008), Sotorasib (AMG510, MCE # HY-114277), MRTX1133 (MCE #HY-134813). Ulixertinib was provided by BioMed Valley under a material transfer agreement (MTA), and Copanlisib and Afatinib were provided by the NCI Cancer Therapy Evaluation Program (CTEP) under MTAs. For all drug treatments, an applicable concentration of 0 (zero) indicates vehicle.

### Lentiviral and retroviral production and transduction

To produce the lentivirus particles, shRNA encoding plasmid (Supplementary Table 2) was mixed with packaging plasmids psPAX2 and pMD2.G in 6 µg:3 µg:1.5 µg ratio in serum free DMEM medium and 4X polyethyleneimine (PEI) transfection reagent (4 µL PEI 1 mg/mL for 1 µg DNA) was added. Similarly, to produce retrovirus particles, equal amount of (4 µg each) of retroviral vector containing gene of interest (Supplementary Table 3) and an envelope plasmid (PCL10A1) were added in serum free DMEM medium and 4X polyethyleneimine (PEI)

transfection reagent (4 µL PEI (1 mg/mL) for 1 µg DNA). After 15 min incubation at room temperature and the mixture was added dropwise onto 293 T cells in 10 cm plates. Next day, medium was replaced with fresh 10% FBS DMEM after 14–16 h of post-transfection. After 48 h and 72 h post-transfection virus was collected and filter by 0.45-micron filter. Target cells were transduced with virus in presence of 10 µg/mL polybrene (Sigma) for 24 h and replaced with fresh medium, then selected with puromycin (2 µg/mL) for 5 days. Surviving polyclonal cells was confirmed by immunoblot analysis of the target proteins before using for desired experiments.

### In vitro cell viability assay and calculation of combination indices

5000 to 10,000 cells/well were plated in triplicates in 96-well plates one day prior to addition of the inhibitors at the indicated final concentrations. After 5 days of culture, viability assay was measured using Resazurin (or Alamar Blue) colorimetric analysis as described[73]. For drug interaction studies, cells were cultured in triplicates in the presence of six fixed-ratio concentrations for 96 h followed by Alamar Blue viability assay. For Supplementary Fig. 2c, concentrations of the drugs were as provided in Supplementary Table 4. Combination indices were calculated using Compusyn software as described[18,74].

### 2D clonogenic assay

Cells were seeded at density 100–200 cells per well (6–12 well format) in regular culture media 24 h before drug treatment. Media was replenished twice weekly. After 3 weeks, cells were washed with PBS and fixed with 4% formaldehyde following the staining with 0.5% crystal violet. Plates were digitally scanned, and colonies were quantified using particle analyzer on ImageJ software.

### Xenograft tumourigenesis assay

Briefly, 2–5 million PDAC cells or 5X5X5mm chunks of cryopreserved PDXs were mixed 1:1 (v/v) with Matrigel matrix (Corning, NY, USA) and inoculated into both flanks of 8 to 12-week-old female NOD-SCIDγ mice (catalog#005557, purchased from the Jackson Laboratory) by needle injection or small incision subcutaneously at the flanks of each mouse. Treatments were initiated when the tumours reached ~100 mm³ in volume. Dosages of each drug: ulixertinib 100 mg/kg BID 5 days/week by oral gavage; afatinib 12.5 mg/kg/day by oral gavage; trametinib 0.5 mg/kg/day by oral gavage; copanlisib 10 mg/kg by tail vein injection over >20 min every other day three times per week when not combined with gemcitabine, or twice weekly when combined with gemcitabine; DS-8201 4 mg/kg by tail vein injection weekly; gemcitabine 75 mg/kg by intraperitoneal injection weekly, MRTX1133 10 mg/kg/day by intraperitoneal injection, AMG-510 10 mg/kg/day by intraperitoneal injection. Mice were euthanized when vehicle-treated tumours reached the maximum allowed volume (~2000 mm³) or when any of the humane endpoints (>20% decrease of body weight from baseline, or from appearance and inactivity) was reached, as described in the IACUC protocol (#22-0101). There were a few occasions, as shown in the Source Data file, when mice with tumors reaching 2000 mm³ that were not immediately sacrificed and euthanasia was delayed till the next measuring day, due to oversight of a mouse technician who measured the tumor diameters without performing the volume calculations. The Division of Comparative Medicine at Washington University was aware of these events, issued warning and corrective actions were taken. No sex and gender analysis were carried out in this study because PDAC affects all genders equally.

### Flow cytometry

For apoptotic assay cancer cells were stained using Annexin V-FITC and propidium iodide (PI, BD bioscience #556547), detected with FACS Calibur (BD bioscience). For surface and total HER2 expression

analysis, cells were stained with Alexa Fluor647 anti-human HER2 (BioLegend). The results were analyzed and quantified using FlowJo software (BD bioscience) using published gating strategy[75] as shown in Supplementary Fig. 9a, b.

## Mass spectrometry

293 T cells were retrovirally infected with pBabepuro-FLAG-KRASG12V or pBabepuro-FLAG-RALAG23V and completed puromycin selection. Three 10 cm plates of each cell lines were grown to 80% confluency prior to harvesting for immunoprecipitation using Pierce™ Anti-DYKDDDDK Magnetic Agarose (cat A36797) according to manufacturer's protocol. The three batches of beads from each cell line were pooled for further processing. Bead-bound proteins were eluted by SDS sample buffer and subjected to filter-aided sample preparation (FASP) and trypsin digestion. Briefly, 30 μl samples were mixed with 200 μl UA buffer consisting of 8 M urea (Sigma, U5128) in 0.1 M Tris-HCl, pH 8.5, and added to Microcon YM-30 filter units (Millipore, MRCF0R030). Samples were spun for 15 min at $14,000 \times g$ and washed twice with 100 μl UA buffer by centrifugation at the same speed for the same length of time. 100 μL of 50 mM iodoacetamide (freshly dissolved in UA buffer) were added, incubated for 20 min at 20 °C in the dark. Samples were centrifuged at $14,000 \times g$ for 10 min, washed twice with 100 μl UA buffer, and 60 μL of sequencing-grade trypsin (Sigma, #11418025001) (200–400 ng total) in 50 mM ammonium bicarbonate was added to the filter units. Following overnight digestion at 37 °C, samples were collected by centrifugation at $14,000 \times g$ for 10 min. 50 μl of 0.5 M NaCl was added to the filters, centrifuged at $14,000 \times g$ for 10 min. Pooled eluates were acidified to 5% formic acid (FA), cleaned up by C18 zip-tips (# ZTC18S096, Millipore, and resuspended in 15 μl 1% formic acid/1% acetonitrile.

Samples were analyzed by reverse-phase liquid chromatography-electrospray ionization-MS/MS using an Eksigent cHiPLC Nanoflex microchip system connected to a quadrupole time-of-flight TripleTOF 5600 mass spectrometer (ABSCIEX). The Nanoflex system uses replaceable microfluidic traps and columns packed with ChromXP C18 (200 um ID x 15 cm, 3 μm particle, 120 Å) for online trapping, desalting, and analytical separations. Solvents composed of water/acetonitrile/formic acid (A, 100/0/0.1%; B, 0/100/0.1%). Peptides were loading onto the column with 98% mobile phase A. After online trapping, peptide mixtures were eluted into analytical column at a flow rate of 800 nL/min using the following gradient: (1) starting at 2% solvent B; (2) 2–5% solvent B from 0 to 12 min; (3) 5–22% solvent B from 12 to 120 min; (4) 22–30% solvent B from 120 to 150 min; (5) 30–80% solvent from 150 to 165 min; and finally 80%(vol/vol) solvent from 165 to 169 min with a total run time of 180 min including mobile phase equilibration. The LC column was maintained at 35 °C during the run. For Data dependent acquisitions, mass spectra and tandem mass spectra were recorded in positive-ion and high-sensitivity mode. The nanospray needle voltage was typically 3800 V. After acquisition of each sample, TOF MS spectra and TOF MS/MS spectra were automatically calibrated during dynamic LC-MS and MS/MS auto calibration acquisitions by injecting 50 fmol β-galactosidase. For collision-induced dissociation tandem MS (CID-MS/MS), the mass window for precursor ion selection of the quadrupole mass analyzer was set to ±1 *m/z*. The precursor ions were fragmented in a collision cell using nitrogen as the collision gas. Advanced information-dependent acquisition (IDA) was used for MS/MS collection on the TripleTOF 5600 to obtain MS/MS spectra for the 20 most abundant parent ions following each survey MS1 scan (allowing typically for 80 ms acquisition time per each MS/MS). Dynamic exclusion features were set to an exclusion mass width of 50 mDa and an exclusion duration of 30 s.

Protein identification and MS1 quantification were performed with MaxQuant (Cox and Mann, 2008) against the Uniprot Human Reference Proteome. The MS/MS spectra were searched with fixed modification of Carbamidomethyl-Cysteine, variable modifications of oxidation (M), acetylation (protein N-term). Search parameters were set to an initial precursor ion tolerance of 0.07 Da, MS/MS tolerance at 40 ppm and requiring strict tryptic specificity with a maximum of two missed cleavages. The minimum required peptide length was set to seven amino acids. Peptide identification FDR was set at 1%.

## Gene set enrichment analysis

Genes in the RNAseq differential expression data were ranked by Log2 fold change, and pre-ranked gene set enrichment analysis was performed using ranked lists. The ranking metric was set to "difference-of-classes" because the expression data were in Log2 units. Otherwise, GSEAv.4 was used for analysis in the default format. The generated data were exported and graphed using the GraphPad Prism v8 software.

## Statistics and reproducibility

All animal experiments were conducted, and tumour volumes measured in blinded fashion by at least two independent lab members after randomization. No statistical method was used to pre-determine sample size. No data were excluded from the analyses All results, when applicable, are expressed as the mean ± SEM (standard error of the mean). Statistical analyses were performed using Prism (v8/9/10) software. Paired or unpaired Student's two-tailed *t*-tests were used to compare two groups when appropriate. For multiple groups, two-way analysis of variance (ANOVA) with Tukey's or Dunnett's post-hoc test was used. Statistical significance was set at $P < 0.05$.

## Reporting summary

Further information on research design is available in the Nature Portfolio Reporting Summary linked to this article.

## Data availability

RNA-seq differential expression data of HPAC *shTRIM21*-knockdown and scramble shRNA cells have been deposited in NCBI's Gene Expression Omnibus (GEO) repository under the GEO series accession number GSE208568. All remaining data can be found in the Article, Supplementary and Source Data files. Source data are provided with this paper.

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

## Acknowledgements

This study was funded by Washington University PDXNet U54CA224083, NIH R37CA219697, NIH R01CA262414, and the Alvin J. Siteman Cancer Center Siteman Investment Program (supported by the Barnard Trust and Foundation for Barnes-Jewish Hospital). We thank BioMed Valley and Bayer (through NCI CTEP) for providing drugs for the study and for reviewing the manuscript. Research reported in this publication was supported by the National Cancer Institute of the National Institutes of Health under Award Number U54CA224083. The content is solely the responsibility of the authors and does not necessarily represent the official views of the National Institutes of Health.

## Author contributions

A.B., P.L. and K.-H.L. conceived the ideas and designed the experiments. A.B., P.L., K.S., Y.C., K.Z., V.S., I.A.K., H-P.C., P.B.D., L.L., Y.G., P.M.G., and J.M.H. performed, analyzed, and interpreted the experiments. C.-K.M., J.M., and M.B.R. analyzed the data. A.B. and K.-H.L. wrote the manuscript. A.B. and P.L. revised the paper. L.D., R.G., S.D., J.M., W.G.H., R.C.F., D.G.D., D.K., A.W.-G. and K.-H.L. provided PDX models, helped secured funding and supervised this study. All authors read and approved the final paper.

## Competing interests

A.W.-G. is currently on sabbatical from Washington University and employed by Jacobio Pharmaceuticals, Inc. Deborah Knoerzer was employed for BioMed Valley Discovery. The remaining authors declare no competing interests.

## Additional information

¹Division of Oncology, Department of Internal Medicine, Washington University School of Medicine, St. Louis, MO 63110, USA. ²Department of General
Surgery, Shengjing Hospital of China Medical University, Shenyang, China. ³Section of Hepatobiliary Surgery, Department of Surgery, Washington University
School of Medicine, St. Louis, MO 63110, USA. ⁴BioMed Valley Discoveries, Kansas City, MO 64111, USA. ⁵Department of Pathology and Immunology,
Washington University School of Medicine, St. Louis, MO 63110, USA. ⁶These authors contributed equally: Ashenafi Bulle, Peng Liu.
✉e-mail: kian-huat.lim@wustl.edu

