## [Peer Review File · Nature Communications]

Reviewers' Comments:

Reviewer #2:

Remarks to the Author:

The main findings in the current version are:

- MAPK inhibition leads to HER2 activation. Mechanism: DUSP6 binds and dephosphorylates HER2. ERK inhibition provokes proteasomal degradation of DUSP6 by loss of interaction with TRIM21 which stabilizes DUSP6 by preventing polyubiquitination. The lack of DUSP6 results in HER2 activation.
- Combining ERK and HER inhibitors slowed, but did not arrest tumor progression of PDXs, while MAPK inhibitors in combination with trastuzumab deruxtecan (DS-8201a), an anti-HER2 antibody conjugated with cytotoxic chemotherapy, led to sustained tumor regression.
- KRAS inhibitors also resulted in HER2 activation, providing a strong rationale for testing the combination of KRAS inhibitors and DS-8201a in PDAC.

The binding of DUSP6 to HER2 is demonstrated in this study by proximity ligation assays and co-immunoprecipitation assays with HER2. The finding of HER2 as a substrate of DUSP6 should still be demonstrated with recombinant DUSP6, available from multiple commercial sources, as this will substantiate a significant mechanistic finding of this work. This finding is important because DUSP6 is selective for ERK1/2 relative to, for example, ERK5, which also possesses a TEY motif (PMID: 18280112). In this regard, DUSP6 has previously been shown to interact with some other proteins including, for example, the progesterone receptor (PMID: 23921636). Terminology in the text involving the TEY motif is misleading and warrants correction. That motif contains the ERK2 activation loop sites dephosphorylated by DUSP6, but is not the DUSP6 binding site on ERK1/2. The binding site is the CD motif on the backside of the kinase domain, clear from multiple crystal structures, and originally identified in a genetic screen in flies (PMID: 12754209). The relevance of DUSP6 in PDAC has been pointed out in previous work (PMID: 15824892), which found that DUSP6 is silenced by promoter methylation in advanced stages of pancreatic cancer. The lack of DUSP6 increasing tumor progression aligns with the findings in this manuscript. In the present work, a detailed mechanism highlighting DUSP6 relevance is provided with a different perspective: the acquired resistance to MAPK pathway inhibitors.

Overall, there are substantial improvements in the manuscript in comparison to its initial version. Most of the concerns have been effectively addressed, misleading data have been excluded, new additional data have been incorporated, confusing experiments have been repeated, and the presentation of the information has been restructured resulting in enhanced coherence and more logical flow. Yet, there are several points that must be addressed:

- 1) In Figure 1h, downregulation of DUSP6 is observed by the overexpression of HER1, HER2 and, especially, HER3. However, in lines 183-184 the authors point out only HER2-DUSP6 negative correlation suggesting a mechanistic link. Unlike the other two isoforms, HER3 is not increasing pERK levels likely due to the lack of HER1 and HER2 expression, what is the mechanistic link between HER3-DUSP6?
- 2) In extended data figure 2a, ulixertinib seems to downregulate DUSP3 apart from DUSP4 and DUSP6. Is there a reason why DUSP3, DUSP5, DUSP8 and DUSP10 are not tested in the pancreatic cell lines (figure 2a) but they are in 293T (extended data figure 2a)? Comparing the 24 h time point in extended data figure 2b with figure 2a Pa01c cells, DUSP4 and DUSP6 downregulation at the same or double ulixertinib concentration used in extended data 2b is not as pronounced as in figure 2b. Why is that difference? In figure 2b, the proteasome inhibitor bortezomib clearly decreases immunoprecipitated DUSP4 in the absence of ulixertinib and, in the case of DUSP6, as much as with ulixertinib alone. However, the degradation of DUSP4 is only observed upon ulixertinib treatment in the total lysates and no difference is observed in the total lysates for DUSP6 whatsoever. What is the reason of this discrepancy? The cell line used in this panel is not indicated in the figure, figure legend or main

text.

Although the prevention of ERK and MEK inhibitor-mediated degradation of DUSP4 and DUSP6 by bortezomib actually, at least partially, prevents HER2 phosphorylation, there is an incongruence in the controls in figure 2c. Bortezomib alone is increasing HER2 Y1248 phosphorylation in Pa01c cells compared to the control, even though to a lesser extent than ulixertinib or trametinib alone. If DUSP4 and DUSP6 expression levels are significantly higher due to the prevention of DUSPs degradation in the presence of the proteasome inhibitor, how is it explained that pHER2 increases (lane 2 compared to lane 1) when the specific HER2 phosphatase, DUSP6, is in a higher concentration? On the contrary a slight decrease in p-HER2 is observed in HPAC cells when treated with bortezomib which makes sense with DUSP6 increase, which is supposed to dephosphorylate HER2. In HPAC cells bortezomib shows an opposite effect in DUSP4 expression, which slightly decreases. Where does this difference between these two cell lines come from? The only visible difference is the effect of bortezomib in DUSP4 expression. Nevertheless, DUSP6 being higher in both cell lines has opposite effects in p-HER2 levels in the presence of proteasome inhibitor. Is this observed in other replicates as well? This is not consistent with what is observed in figure 2d in Pa01c cells, where the downregulation of DUSP6 increases phosphorylation in Y1248 of HER2 and in figure 2e where overexpression of DUSP6 decreases this phosphorylation.

As claimed by reviewer 1, in figure 2k, "DUSP6 didn't decrease despite the enhanced polyubiquitination". The authors' explanation is that "a longer treatment (≥ 4 hours) with trametinib or ulixertinib leads to significant decline of DUSP6, making it difficult to appreciate the relationship between DUSP6 and HER2. In figure 2k, the experiment was performed after ~ 6 hours of treatment and all cells co-treated with bortezomib to prevent DUSP6 downregulation. This allows visualization of DUSP6 polyubiquitination in each condition when the total DUSP6 levels are comparable". In other experiments cells are treated overnight with ulixertinib and trametinib and then 6 h with bortezomib. If the length of treatment can give rise to so many variations, should not the 6 h treatment rationale be followed throughout the whole manuscript?

3) Overall, a lot of variability is observed in treatment conditions between experiments from 3 h to 6 h, overnight or 24 h with ulixertinib and trametinib. For example, in figure 3a the inhibitors are applied overnight in 293T whereas in figure 3h only for 3 h in the same cell line. The rationale for this variability should be indicated in text.

The point that the authors want to make with figure 3a is not clear. As shown by KRAS overexpression or downregulation in figure 3b, c and d the presence of mutant KRAS increases DUSP6 stability. Yet ulixertinib treatment provokes DUSP6 degradation in Pa01c and MIA Paca-2 after 6 h of treatment, however in 293T DUSP6 is upregulated when KRASG12V is overexpressed (fig 3a) and the treatment with ulixertinib or trametinib does not decrease DUSP6 whatsoever even after overnight treatment.

A conclusion made from figure 3 is that ulixertinib and trametinib do not have the same effect in DUSP6 expression in 293T as in PDAC cells. For this reason, the interaction assays done in 293T may not be applicable to pancreatic cells where probably the downregulation of DUSP6 due to the inhibitors might alter the results of the interaction assay.

Minor:

- The alignment of the receptor and ERK1/2 activation loops is shifted. The conserved glutamate in the APE motif of ERK and in the ALE sequence in the receptors should be aligned with each other.
- Figure 1a and extended data figure 1a and 1b legends: Gemcitabine is missing the first i. In 1b and 1d "ithe" should be "the". In extended data figure 1h legend "toral" should be "total".
- Line 148: "an MEK inhibitor" should be "a MEK inhibitor".
- Line 240: "pr" should be "or".
- Line 708: extended data figure legend 2j μ symbol is duplicated.
- Line 278: Fig. 4j should be 3j
- Figure 4c legend: trametinib dosage is indicated (line 769) but it was not used in the experiment.
- Extended data figure 4 (supporting figure 6) should be supporting figure 4. Also, first "from" in the figure legend should be removed.
- Line 352: typo, "shemistry" should be "chemistry".
- Line 380: missing reference to figure 6f.
- Typo in line 416 "progression-free survival" is repeated twice.
- Typo in line 434 "bone fide" should be "bona fide".
- Typo in line 446 "an MEK inhibitor" should be "a MEK inhibitor".
- In several places the word 'specific' is used and should be changed to selective.

In summary, the main issue with this strategy is that, as pointed out by the authors, DS-8201a does not recognize murine HER2. For this reason, the toxicity assays in mice do not guarantee safety in patients. The clinical trials already accepted and planned through the NCI will be crucial to determine the actual success of this promising strategy.

Reviewer #3:

Remarks to the Author:

The authors have made several improvements to the manuscript during the revision process. The overall flow of the manuscript is improved with the merging of Figures 1 and 2 and the reduced focus of the induction of the MAPK pathway under stress. The key experiments from Figures 1-3 have been repeated and associated Western blots are much less noisy and easier to interpret. Moreover, several key experiments have been added including (1) showing that DUSP6 binds directly to HER2 and ERK through a TEY motif (2) showing that novel small molecule inhibitors of KRAS can also upregulate p-HER2 and total HER2 and that there is pre-clinical efficacy when combined with trastuzumab deruxtecan. The manuscript provides rationale for testing this combination and is appropriate for publication in Nature Communications in its current format. I do not recommend additional experimental revisions.

Minor Comments:

- The use of copanlisib in Figure 4 is presented without clear rationale and is the first use of PI3K inhibitor in this paper. As is, it is confusing and detracts from the flow of the text. Though there is rationale for PI3K inhibitors and MEK inhibitors, this should be explained clearly.

Point-by-point response to Reviewers' comments

Reviewer #2:

The main findings in the current version are:

- MAPK inhibition leads to HER2 activation. Mechanism: DUSP6 binds and dephosphorylates HER2. ERK inhibition provokes proteasomal degradation of DUSP6 by loss of interaction with TRIM21 which stabilizes DUSP6 by preventing polyubiquitination. The lack of DUSP6 results in HER2 activation.
- Combining ERK and HER inhibitors slowed, but did not arrest tumor progression of PDXs, while MAPK inhibitors in combination with trastuzumab deruxtecan (DS-8201a), an anti-HER2 antibody conjugated with cytotoxic chemotherapy, led to sustained tumor regression.
- KRAS inhibitors also resulted in HER2 activation, providing a strong rationale for testing the combination of KRAS inhibitors and DS-8201a in PDAC.

The binding of DUSP6 to HER2 is demonstrated in this study by proximity ligation assays and co-immunoprecipitation assays with HER2. The finding of HER2 as a substrate of DUSP6 should still be demonstrated with recombinant DUSP6, available from multiple commercial sources, as this will substantiate a significant mechanistic finding of this work. This finding is important because DUSP6 is selective for ERK1/2 relative to, for example, ERK5, which also possesses a TEY motif (PMID: 18280112). In this regard, DUSP6 has previously been shown to interact with some other proteins including, for example, the progesterone receptor (PMID: 23921636). Terminology in the text involving the TEY motif is misleading and warrants correction. That motif contains the ERK2 activation loop sites dephosphorylated by DUSP6 but is not the DUSP6 binding site on ERK1/2. The binding site is the CD motif on the backside of the kinase domain, clear from multiple crystal structures, and originally identified in a genetic screen in flies (PMID: 12754209).

Response: We appreciate the reviewers for pointing this out and agree. Along the similar line of work done by the Zhang lab (which published PMID: 12754209 as the Reviewer pointed out), phosphorylation of T185 and Y187 in the TEY motif of ERK2 enhances the binding of its CD motif to DUSP6 by 6-fold using hydrogen/deuterium exchange mass spectrometry (PMID: 17046812), indicating the TEY motif is indeed involved in promoting ERK2-DUSP6 binding. This binding then resulted in stepwise dephosphorylation of T185 and Y187 of ERK2 by DUSP6. Our finding that mutation of the TEY motif in HER2 protein lowers its interaction with DUSP6 suggests that similar mechanism may be at play, although the binding interfaces between HER2 and DUSP6 have not been molecularly resolved. We have corrected this aspect in our revised manuscript (line 227-238).

The relevance of DUSP6 in PDAC has been pointed out in previous work (PMID: 15824892), which found that DUSP6 is silenced by promoter methylation in advanced stages of pancreatic cancer. The lack of DUSP6 increasing tumor progression aligns with the findings in this manuscript. In the present work, a detailed mechanism highlighting DUSP6 relevance is provided with a different perspective: the acquired

resistance to MAPK pathway inhibitors. Overall, there are substantial improvements in the manuscript in comparison to its initial version. Most of the concerns have been effectively addressed, misleading data have been excluded, new additional data have been incorporated, confusing experiments have been repeated, and the presentation of the information has been restructured resulting in enhanced coherence and more logical flow. Yet, there are several points that must be addressed:

1) In Figure 1h, downregulation of DUSP6 is observed by the overexpression of HER1, HER2 and, especially, HER3. However, in lines 183-184 the authors point out only HER2-DUSP6 negative correlation suggesting a mechanistic link. Unlike the other two isoforms, HER3 is not increasing pERK levels likely due to the lack of HER1 and HER2 expression, what is the mechanistic link between HER3-DUSP6?

Response: We repeated this experiment using HER1, HER2, HER2 alone (**Fig. 1h**) or HER1/2, HER1/3 and HER2/3 (**Extended Fig. 2d**) to delineate the impact of each HER family member alone or as heterodimer, on DUSP4 and DUSP6. In both settings, DUSP6 was downregulated only in the presence of HER2 (either alone or when co-expressed with HER1 or HER3). These results support the established literature that HER3 requires other HER members especially HER2 for signaling, and to impact DUSP6 expression.

2) In extended data figure 2a, ulixertinib seems to downregulate DUSP3 apart from DUSP4 and DUSP6. Is there a reason why DUSP3, DUSP5, DUSP8 and DUSP10 are not tested in the pancreatic cell lines (figure 2a) but they are in 293T (extended data figure 2a)?

Response: We repeated this experiment and showed all the other DUSPs in the PDAC cell lines (**Fig. 2a**). Only DUSP4 and DUSP6 are downregulated following treatment with MEK or ERK inhibitors.

Comparing the 24 h time point in extended data figure 2b with figure 2a Pa01c cells, DUSP4 and DUSP6 downregulation at the same or double ulixertinib concentration used in extended data 2b is not as pronounced as in figure 2b. Why is that difference?

Response: We repeated this experiment using newly prepared ulixertinib three independent times and now are able to show that ulixertinib is able to time-dependently downregulate DUSP4 and DUSP6 at 1 and 2uM. We provided quantitative analysis as well (**Extended Data Fig. 2c**).

In figure 2b, the proteasome inhibitor bortezomib clearly decreases immunoprecipitated DUSP4 in the absence of ulixertinib and, in the case of DUSP6, as much as with ulixertinib alone. However, the degradation of DUSP4 is only observed upon ulixertinib treatment in the total lysates and no difference is observed in the total lysates for DUSP6 whatsoever. What is the reason of this discrepancy? The cell line used in this panel is not indicated in the figure, figure legend or main text.

Response: We provided details on this cell line which is 293T cells stably expressing empty vector or FLAG-tagged DUSP4 or DUSP6. We repeated this western blot using the same lysates and provided quantification that ulixertinib decreased expression of both FLAG-tagged DUSP4 and -DUSP6 but this decrease could be reversed by co-treatment with bortezomib (**Fig. 2b**).

Although the prevention of ERK and MEK inhibitor-mediated degradation of DUSP4 and DUSP6 by bortezomib actually, at least partially, prevents HER2 phosphorylation, there is an incongruence in the controls in figure 2c. Bortezomib alone is increasing HER2 Y1248 phosphorylation in Pa01c cells compared to the control, even though to a lesser extent than ulixertinib or trametinib alone. If DUSP4 and DUSP6 expression levels are significantly higher due to the prevention of DUSPs degradation in the presence of the proteasome inhibitor, how is it explained that pHER2 increases (lane 2 compared to lane 1) when the specific HER2 phosphatase, DUSP6, is in a higher concentration? On the contrary a slight decrease in p-HER2 is observed in HPAC cells when treated with bortezomib which makes sense with DUSP6 increase, which is supposed to dephosphorylate HER2. In HPAC cells bortezomib shows an opposite effect in DUSP4 expression, which slightly decreases. Where does this difference between these two cell lines come from? The only visible difference is the effect of bortezomib in DUSP4 expression. Nevertheless, DUSP6 being higher in both cell lines has opposite effects in p-HER2 levels in the presence of proteasome inhibitor. Is this observed in other replicates as well? This is not consistent with what is observed in figure 2d in Pa01c cells, where the downregulation of DUSP6 increases phosphorylation in Y1248 of HER2 and in figure 2e where overexpression of DUSP6 decreases this phosphorylation.

Response: Response: We thank the Reviewer for pointing these out. The previous issue was due to suboptimal blot transfer or exposure as lane 1 of Pa01c cells was at the edge of the gel. We re-ran the lysates from this experiment and now show that higher level of DUSP6 resulting from bortezomib coincided with lower p-HER2 levels in both Pa01c and HPAC cell lines (**Fig. 2c**).

As claimed by reviewer 1, in figure 2k, “DUSP6 didn’t decrease despite the enhanced polyubiquitination”. The authors’ explanation is that “a longer treatment (≥ 4 hours) with trametinib or ulixertinib leads to significant decline of DUSP6, making it difficult to appreciate the relationship between DUSP6 and HER2. In figure 2k, the experiment was performed after ~6 hours of treatment and all cells co-treated with bortezomib to prevent DUSP6 downregulation. This allows visualization of DUSP6 polyubiquitination in each condition when the total DUSP6 levels are comparable”. In other experiments cells are treated overnight with ulixertinib and trametinib and then 6 h with bortezomib. If the length of treatment can give rise to so many variations, should not the 6 h treatment rationale be followed throughout the whole manuscript?

Response: We are terribly sorry for this mistake. **Figure 2k** was performed with 16 hours of trametinib or ulixertinib treatment (not 6 hours), as with most other experiments. We did show that DUSP4 and DUSP6 levels decrease after 6 hours of treatment (**Extended Fig. 2c**). By treating the cells with bortezomib 4 hours before treatment, we are able to protect HA-tagged DUSP6 from being degraded, which allows us to perform IP using HA-beads in equal quantity in each lane. Only by doing this are we able to discern an increase interaction between DUSP6 and HER2 following ulixertinib or trametinib treatment (**Fig. 2k**).

3) Overall, a lot of variability is observed in treatment conditions between experiments from 3 h to 6 h, overnight or 24 h with ulixertinib and trametinib. For example, in figure 3a the inhibitors are applied overnight in 293T whereas in figure 3h only for 3 h in the same cell line. The rationale for this variability should be indicated in text.

Response: We apologize for this issue and appreciate the suggestion. To be consistent, we have repeated **Fig. 3h** with overnight treatment just to be consistent with other data. All in vitro drug experiments in this manuscript are now done overnight (~16 hours) or 24 hours, and we clearly indicated these in figure legends. The only experiment that was done at 4 hours is **Fig. 2j** because we are trying to visualize and quantify the interaction between endogenous DUSP6 and HER2 as an early event following drug treatment. As shown in **Extended Data Fig. 2c**, treatment with either inhibitor for 6 hours results in significant downregulation of DUSP6, which will skew our results.

The point that the authors want to make with figure 3a is not clear. As shown by KRAS overexpression or downregulation in figure 3b, c and d the presence of mutant KRAS increases DUSP6 stability. Yet ulixertinib treatment provokes DUSP6 degradation in Pa01c and MIA Paca-2 after 6 h of treatment, however in 293T DUSP6 is upregulated when KRASG12V is overexpressed (fig 3a) and the treatment with ulixertinib or trametinib does not decrease DUSP6 whatsoever even after overnight treatment. A conclusion made from figure 3 is that ulixertinib and trametinib do not have the same effect in DUSP6 expression in 293T as in PDAC cells. For this reason, the interaction assays done in 293T may not be applicable to pancreatic cells where probably the downregulation of DUSP6 due to the inhibitors might alter the results of the interaction assay.

Response: We appreciate the Reviewer's comment. We have since repeated **Fig. 3a** three times and show that overnight treatment with trametinib and ulixertinib did decrease DUSP6 protein level in 293T cells expressing KRASG12V. Compared to 293T vector control cells, the half-life of endogenous DUSP6 is higher when KRASG12V is present (**Fig. 3b**), indicating that KRAS oncoprotein positively regulates DUSP6 stability. We acknowledge the Reviewer's comments on 293T vs PDAC cells. Therefore, to our best ability, we silenced *KRAS* or used different KRAS inhibitors in PDAC cells to corroborate findings in 293T cells in **Fig. 3**.

Minor:

- The alignment of the receptor and ERK1/2 activation loops is shifted. The conserved glutamate in the APE motif of ERK and in the ALE sequence in the receptors should be aligned with each other.

Response: Thanks so much as we corrected this (**Fig. 2g**).

- Figure 1a and extended data figure 1a and 1b legends: Gemcitabine is missing the first i. In 1b and 1d "ithe" should be "the". In extended data figure 1h legend "toral" should be "total".

- Line 148: "an MEK inhibitor" should be "a MEK inhibitor".

- Line 240: "pr" should be "or".

- Line 708: extended data figure legend 2j μ symbol is duplicated.

- Line 278: Fig. 4j should be 3j

- Figure 4c legend: trametinib dosage is indicated (line769) but it was not used in the experiment.

- Extended data figure 4 (supporting figure 6) should be supporting figure 4. Also, first "from" in the figure legend should be removed.

- Line 352: typo, "shemistry" should be "chemistry".

- Line 380: missing reference to figure 6f.

- Typo in line 416 "progression-free survival" is repeated twice.

- Typo in line 434 "bone fide" should be "bona fide".

- Typo in line 446 "an MEK inhibitor" should be "a MEK inhibitor".

- In several places the word 'specific' is used and should be changed to selective.

Response: Thanks so much. We are terribly sorry and corrected these.

In summary, the main issue with this strategy is that, as pointed out by the authors, DS-8201a does not recognize murine HER2. For this reason, the toxicity assays in mice do not guarantee safety in patients. The clinical trials already accepted and planned through the NCI will be crucial to determine the actual success of this promising strategy.

Reviewer #3

The authors have made several improvements to the manuscript during the revision process. The overall flow of the manuscript is improved with the merging of Figures 1 and 2 and the reduced focus of the induction of the MAPK pathway under stress. The key experiments from Figures 1-3 have been repeated and associated Western blots are much less noisy and easier to interpret. Moreover, several key experiments have been added including (1) showing that DUSP6 binds directly to HER2 and ERK through a TEY motif (2) showing that novel small molecule inhibitors of KRAS can also upregulate p-HER2 and total HER2 and that there is pre-clinical efficacy when combined with trastuzumab deruxtecan. The

manuscript provides rationale for testing this combination and is appropriate for publication in Nature Communications in its current format. I do not recommend additional experimental revisions.

Minor Comments: The use of copanlisib in Figure 4 is presented without clear rationale and is the first use of PI3K inhibitor in this paper. As is, it is confusing and detracts from the flow of the text. Though there is rationale for PI3K inhibitors and MEK inhibitors, this should be explained clearly.

Response: We appreciate this comment. In **Fig. 1i** we show that both trametinib and ulixertinib can upregulate p-AKT (S473), which is a well-established adaptive survival event in PDAC. Co-targeting the MAPK and PI3K pathways is reported in numerous literature and we utilize copanlisib as a control to compare its efficacy with afatinib. We explained this in line 290.

Reviewers' Comments:

Reviewer #2:

Remarks to the Author:

Having carefully reviewed the revised manuscript, I confirm that all my concerns and suggestions have been effectively addressed. The repetition of certain experiments, initially conducted under inconsistent conditions or yielding confusing results, along with the incorporation of suggested experiments concerning HER2-DUSP6 interaction, has significantly enhanced the robustness of the study. In my opinion, the paper is ready for publication in NCOMMS now.